# Joint Adaptation of Uni-modal Foundation Models for Multi-modal Alzheimer's Disease Diagnosis

**Wentao Gu**[1,2*]   **Yuquan Li**[1*]   **Xinyang Jiang**[2†]   **Zilong Wang**[2]   **Dongsheng Li**[2]
**Zehui Li**[2]   **Zijian Dong**[3]   **Cairong Zhao**[1†]

[1]School of Computer Science and Technology, Tongji University
[2]Microsoft Research Asia
[3]National University of Singapore

## Abstract

Alzheimer's Disease (AD) is a progressive neurodegenerative disorder and a leading cause of dementia worldwide. Accurate diagnosis requires integrating diverse patient data modalities. With the rapid advancement of foundation models in neurobiology and medicine, integrating foundation models from various modalities has emerged as a promising yet underexplored direction for multi-modal AD diagnosis. A central challenge is enabling effective interaction among these models without disrupting the robust, modality-specific representations learned from large-scale pretraining. To address this, we propose a novel multi-modal framework for AD diagnosis that enables joint interaction among uni-modal foundation models through modality-anchored interaction. In this framework, one modality and its corresponding foundation model are designated as an anchor, while the remaining modalities serve as auxiliary sources of complementary information. To preserve the pre-trained representation space of the anchor model, we propose modality-aware Q-formers that selectively map auxiliary modality features into the anchor model's feature space, enabling the anchor model to jointly process its own features together with the seamlessly integrated auxiliary features. We evaluate our method on AD diagnosis and progression prediction across four modalities: sMRI, fMRI, clinical records, and genetic data. Our framework consistently outperforms prior methods in two modality settings, and further demonstrates strong generalization to external datasets and other neurodegenerative diseases such as Parkinson's disease.

## 1 Introduction

Alzheimer's disease (AD) (Li et al., 2022b; Cahill, 2020; Ning et al., 2021; Association, 2019; Dong et al., 2024) is one of the most common causes of dementia worldwide, leading to a progressive decline in cognitive function that significantly interferes with daily activities (Xue et al., 2024; Qiu et al., 2022) among the elderly. With the growing aging population, there is a pressing need for more precise early detection and progression prediction of Alzheimer's disease.

While deep learning has advanced AD analysis, most existing studies focus on developing models on uni-modal data (Liu et al., 2024), such as medical imaging (Wang et al., 2023) or cognitive assessment scores (Fang et al., 2024). However, Alzheimer's disease is a complex neurodegenerative disorder involving diverse and interacting pathological mechanisms, reflected across multiple data modalities. Structural MRI (sMRI) highlights brain atrophy (Ferreira & Busatto, 2011), functional MRI (fMRI) captures neural activity (Dong et al., 2024), non-imaging data (including demographics and clinical assessments) reflect overall patient status (Li et al., 2024a), and genetic data reveal hereditary risks (Beebe-Wang et al., 2021). Recent AD guidelines from NIA-AA also emphasize the necessity of integrating biomarkers from multiple modalities, as each provides complementary

---

*Equal contribution. Work done during Wentao's internship at MSRA.
†Corresponding Authors. Email: xinyangjiang@microsoft.com, zhaocairong@tongji.edu.cn

and valuable insights (Jack Jr et al., 2024). Therefore, integrating complementary information from multiple modalities is essential for a comprehensive understanding and accurate prediction of AD.

While conventional multi-modal approaches have predominantly relied on training models from scratch (Xue et al., 2024; Qiu et al., 2022; Feng et al., 2023a), deep learning is undergoing a paradigm shift toward leveraging large-scale, pre-trained foundation models for downstream tasks adaptation. This approach is particularly advantageous in medical domains like AD, where labeled data are scarce and a more efficient and robust learning method is required. A range of uni-modal foundation models in neurobiology and medicine show strong potential for enhancing AD diagnosis, such as BrainMVP (Rui et al., 2025) and Brain-JEPA (Dong et al., 2024) for brain imaging, large language models for clinical records, and gene foundation models (Dalla-Torre et al., 2025) for genomics, all demonstrating strong performance in their respective domains.

Despite the availability of powerful uni-modal foundation models, integrating them into a unified multi-modal AD diagnosis framework remains a significant challenge. The core difficulty lies in effectively enabling meaningful interaction among foundation models, which requires aligning their feature spaces and integrating their outputs to leverage complementary information. Since each foundation model is pre-trained to capture distinct, modality-specific features, their representations are inherently heterogeneous and well-structured. Naively aligning or merging these spaces may compromise their integrity and reduce their effectiveness.

As a result, our goal is to strike a balance between enabling sufficient interaction among foundation models and preserving the integrity of their pre-trained feature spaces. To address this, we propose a unified multi-modal framework for Alzheimer's Disease diagnosis based on **modality-anchored foundation model interaction**. Specifically, we designate one modality's foundation model as an anchor and freeze most of its parameters to preserve its feature space, while projecting auxiliary modalities' features extracted by other foundation models into this space for cross-modal interaction. This alignment is achieved by our **Modality-aware Q-formers** (Tong et al., 2024; Zong et al., 2024; Alayrac et al., 2022; Liu et al., 2023a), which use learned queries to selectively extract relevant information from the auxiliary modalities and project it to anchor model feature space, enabling the anchor foundation model to jointly process them with the anchor features. Modality-anchored interaction is applied to each modality in turn, and final predictions are aggregated, allowing us to retain the strengths of each foundation model while enabling effective multi-modal integration.

In experiments, the proposed method is evaluated on AD diagnosis and progression prediction tasks involving the four most common data modalities (i.e., sMRI, fMRI, clinical records, and genetic data). By integrating four uni-modal foundation models, our method achieves state-of-the-art performance under both modality-complete and modality-incomplete scenarios from the ADNI dataset (Mueller et al., 2005). We further evaluate our approach on the external OASIS (LaMontagne et al., 2019) and PPMI (Marek et al., 2011) datasets, where it achieves state-of-the-art performance and shows strong generalization to both out-of-distribution AD diagnosis tasks and other neurodegenerative diseases.

## 2 RELATED WORKS

**Multi-modal Fusion Methods for AD Diagnosis.** Alzheimer's Disease (AD) is a complex neuro disorder, and its accurate diagnosis requires multi-modal data integration. Early efforts focus on neuroimaging data combination. Modalities such as MRI, PET, fMRI, and DTI are integrated via methods including shared representations (Ning et al., 2021), GCNs (Song et al., 2022), and 3D networks (Qiu et al., 2024). Subsequent studies integrated non-imaging data, for instance, by using LLMs or deep learning to combine MRI with cognitive scores (Hett et al., 2021; Feng et al., 2023b; Qiu et al., 2022; Chen & Hong, 2024; Xue et al., 2024). Methodologies also evolved to address data challenges like missing modalities (Liu et al., 2023c) or limited labeled samples (Feng et al., 2023a). While prior methods often relied on a limited subset of modalities, our framework is the first to incorporate all three major types of AD-related data: genetic, neuroimaging, and clinical. Our broader modality coverage, combined with the adaptation of foundation models, enables a more comprehensive understanding of AD pathology and improves diagnostic accuracy.

**Adaptation Methods on Foundation Models.** The development of foundation models has significantly impacted healthcare by enabling powerful uni-modal data analysis. In medical imaging, diverse foundation models are utilized for imaging analysis (Wang et al., 2023; Rui et al., 2025; Caro

et al., 2023; Dong et al., 2024), and shows impressive performance in downstream tasks. In genomics, foundation models pre-trained on DNA sequences (Dalla-Torre et al., 2025; Nguyen et al., 2023; Zhou et al., 2023) have shown considerable success in cross-species genomic modeling and analysis. In clinical records, previous efforts (Ben Shoham & Rappoport, 2024; Singhal et al., 2023; Li et al., 2024a) either focus on scaling predictors or attempting to adapt LLMs to clinical predictions. While powerful uni-modal foundation models exist, how to effectively leverage them remains underexplored. Efforts (Zhang et al., 2023b) like M4Survive (Lee et al., 2025) try to integrate medical foundation models using symmetric late-fusion, which may hinder deep inter-modal interactions. In contrast, our work targets more effective interaction across various uni-modal foundation models.

**Q-formers in Multi-modal Pretrained Models** Prior studies primarily used query transformers (q-formers) Li et al. (2022a) or connectors Liu et al. (2023a) to project non-text modalities, such as images, video, or audio, into the text embedding space of large language models (LLMs) for multimodal alignment. For example, BLIP-2 Li et al. (2023) and MiniGPT-4 Zhu et al. (2023) use query transformer to extract features from image patches and output query embeddings that the LLM consumes. InstructBLIP Dai et al. (2023) extend this approach to fuse images, video, and audio, with separate query transformers for each modality projecting their features into the LLM text space. Similarly, speech-, video-, and audio-visual models like EmoQ Yang & Mak (2025), Video-LLaMA Zhang et al. (2023a), and MMS-LLaMA Yeo et al. (2025) employ query transformer to compress their own modality embeddings into textual representations for LLM processing. Differing from those prior works, where query transformers project each modality exclusively into the text embedding space of LLMs, our modality-anchored interaction sequentially treats each modality as the anchor, with the remaining three modalities serving as auxiliary modalities. Consequently, our Q-former is designed to be more general, capable of projecting into any of the four modality spaces when designated as the anchor, rather than being restricted to text.

## 3 METHOD

In this section, we present the methodology of our proposed multi-modal framework for Alzheimer's Disease diagnosis. The overall pipeline to train a multi-modal AD diagnosis model contains two stages. The first stage, uni-modal foundation model adaptation described in Section 3.1, individually fine-tunes each foundation model on its respective modality data to extract highly expressive, modality-specific features for the AD diagnosis tasks. In the second stage, given the uni-modal AD diagnosis models obtained in the first stage, a Modality-anchored Foundation Models Interaction Strategy (elaborated in section 3.2) is adopted to enable interaction among uni-modal models without compromising the integrity of feature space from each model. This is achieved by aligning the feature of auxiliary modalities to the primary modality feature space with Modality-aware Q-formers (elaborated in section 3.3), designed to learn a set of learnable queries to extract relevant information from the auxiliary modalities. Finally, our method gives the final AD diagnosis predictions by combining the outputs of the fine-tuned modality-specific foundation model.

### 3.1 UNI-MODAL FOUNDATION MODEL ADAPTATION

In the first stage, the primary goal is to adapt each uni-modal foundation model to AD diagnosis using limited labeled data from its corresponding modality, leveraging the model's inherent strong representations.

**Problem formulation** Our objective is to predict an individual's Alzheimer's disease status or prodromal progression patterns using multi-modal inputs. In this paper, we focus on four data modalities denoted as $m \in \mathcal{M} = \{\mathtt{s}, \mathtt{f}, \mathtt{c}, \mathtt{g}\}$, where $\mathtt{s}$ and $\mathtt{f}$ refer to neuroimaging data sMRI and fMRI, $\mathtt{c}$ refers to clinical records and $\mathtt{g}$ refers to genetic data. As depicted in stage 1 in Fig 1, given a training dataset $\mathcal{D}_m = \{(x_i^m, y_i)\}_{i=1}^{N_m}$ for modality $m$, where $x_i^m$ is the input sample, $y_i$ is the corresponding diagnosis label, $N_m$ is the size of dataset for modality $m$. All uni-modal foundation models used for AD diagnosis are based on transformer architectures. Each model $F_m(\cdot; \theta_m)$ is attached with a linear classification head that takes the output class token from the transformer predicts the diagnostic label and is fine-tuned individually using the standard cross-entropy loss:

$$\mathcal{L}_m = \frac{1}{N_m} \sum_{i=1}^{N_m} \mathcal{L}_{\text{CE}} \left( F_m(x_i^m; \theta_m), y_i \right). \tag{1}$$

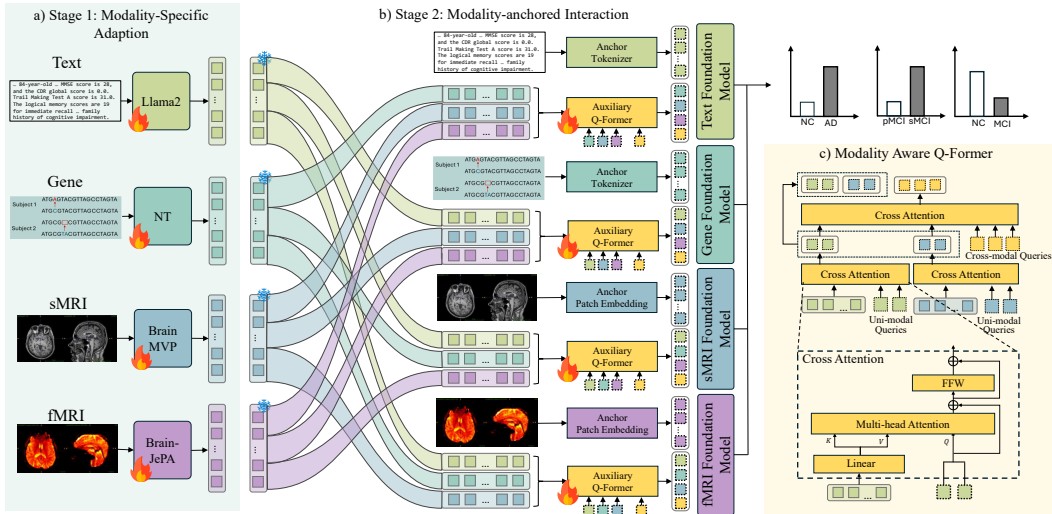

Figure 1: The overall pipeline of the proposed multi-modal AD diagnosis framework. a) In the first stage, the uni-modal foundation model is finetuned to AD diagnosis using limited labeled data from its corresponding modality. b) In the second stage, four uni-modal AD diagnosis models from the first stage are further finetuned to enable cross-modality interactions by modality-anchored interaction. c) Modality-aware Q-former (Auxiliary Q-Former in stage 2) is a transformer-based connector that selectively projects the features from the auxiliary modality to the feature space of the anchor model.

Next, we detail the process of adapting each foundation model and its corresponding uni-modal data for the AD diagnosis task, including the choice of foundation model and the preprocessing steps applied to each input modality, as illustrated in Stage 1 of Fig 1.

**Neuro-image Modalities** Neural-image modalities include structural MRI and functional MRI data. For structural MRI, we adopt BrainMVP (Rui et al., 2025), a large-scale medical imaging foundation model trained on volumetric data with rich anatomical priors. For functional MRI, we employ Brain-JEPA (Dong et al., 2024), which learns generalizable representations of brain dynamics through predictive learning across fMRI datasets.

**Gene Modality** For genetic data, we adopt the NT Transformer (Dalla-Torre et al., 2025), a genomic foundation model pre-trained on nucleotide sequences to support tasks such as disease association and phenotype prediction. The genetic training dataset is constructed with whole genome sequencing (WGS) studies for AD, which provides base-pair level coverage of the entire genome, allowing for a comprehensive assessment of individual genetic variation. We provide full details of variant selection for genetic input and sequence representation constructing pipeline are provided in the Appendix A.2.

**Textual Modality** The textual modality encompasses subjects' comprehensive clinical records, including demographic information, cognitive assessments, neuropsychiatric symptoms, functional abilities, and medical history. To model this modality, we employ LLaMA-2 (Touvron et al., 2023), a large language model capable of capturing latent patterns in both textual and tabular clinical features. We adopt a text template strategy to convert clinical records into description and task prompts. The converting pipeline is detailed in the Appendix A.1.

## 3.2 Modality-anchored Foundation Models Interaction

As shown in Fig 1, in the second stage of the multi-modal training pipeline, we fine-tune the four uni-modal AD diagnosis models from the previous stage introduced in section 3.1 to enable modality-anchored foundation model interaction.

Instead of allowing all uni-modal models to interact equally, the modality-anchored interaction designates one modality as the anchor modality along with its corresponding anchor model obtained

from stage 1. For a specific anchor modality $\hat{m}$, its auxiliary modalities refer to all remaining modalities whose features will be aligned to the anchor modality's feature space, denoted as $\mathcal{M}' = \{m \in \mathcal{M} | m \neq \hat{m}\}$. Given a set of input signals, including anchor modality input $x^{\hat{m}}$ and auxiliary modality inputs $\{X^m | m \in \mathcal{M}'\}$, we first extract features from the auxiliary inputs using their corresponding uni-modal AD diagnosis models obtained from Stage 1. The extracted features from the auxiliary modalities are then aligned to the feature space of the anchor model $F_{\hat{m}}$ using the proposed modality-aware Q-former in section 3.3. Specifically, the aligned auxiliary representation $H^a$ is computed as follows:

$$H^a = \text{Qformer}\big(\text{Concat}(\{F_m(X^m)\}_{m \in \mathcal{M}'})\big), \tag{2}$$

where $\text{Concat}(\cdot)$ denotes the concatenation of the input features vectors, $\text{Qformer}(\cdot)$ refers to the modality-aware Q-former elaborated in the next sub-section 3.3. Then the interaction is achieved by feeding the aligned auxiliary modality tokens $H^a$ into the anchor model $F_{\hat{m}}$ alongside with the anchor modality input to produce the diagnosis prediction. Finally, the anchor model $F_{\hat{m}}$ is further fine-tuned with a standard cross-entropy loss:

$$\mathcal{L}_{\hat{m}} = \frac{1}{N_{\hat{m}}} \sum_{i=1}^{N_{\hat{m}}} \mathcal{L}_{\text{CE}}\left(F_{\hat{m}}\big(\text{Concat}(X^{\hat{m}}, H^a)\big)\right). \tag{3}$$

Building on the modality-anchored interaction described above, we fine-tune each uni-modal model $F_m$ from Stage 1 by designating it as the anchor modality, while treating the remaining modalities as auxiliary inputs. To better preserve the original feature space of the anchor model, we apply LoRA (Hu et al., 2022) fine-tuning, where only a small subset of parameters is updated. The final diagnostic prediction is obtained by aggregating the outputs from all fine-tuned models.

## 3.3 MODALITY-AWARE Q-FORMERS

As shown in Equation 2, to allow effective interaction between the anchor model and the auxiliary models, a transformer-based connector is proposed to selectively project features from the auxiliary modality to the feature space of the anchor model, called modality-aware Q-former. As illustrated in Fig 1, our modality-aware Q-former incorporates two types of information, namely uni-modal and cross modal information.

**Uni-modal Q-formers** Modality-aware Q-former first extracts the uni-modality information from a specific auxiliary modality $m \in \mathcal{M}'$. Specifically, we create a set of learnable tokens to serve as uni-modality queries, denoted as $X_{uq} \in \mathbb{R}^{N_q \times C}$. Given the auxiliary features extracted from the corresponding auxiliary model $F_m$, we first project them to the same dimension as the anchor modality:

$$Z^m = \text{Linear}\big(F_m(X^m)\big) \in \mathbb{R}^{L^m \times C}. \tag{4}$$

Then, the learnable uni-modal queries interact with the projected features through a cross-attention layer, which further projects the auxiliary modality features into the anchor feature space and extracts information relevant to the anchor modality from the auxiliary one $m$:

$$\hat{X}^m = \text{CrossAttn}(Q = X_{uq}^m, K = Z^m, V = Z^m) \tag{5}$$

The resulting output $\hat{X}^m \in \mathbb{R}^{N_q \times C}$ are features containing uni-modal information from auxiliary modality $m$.

**Cross-modal Q-former** Besides uni-modal information, we further propose a set of cross-modal queries $X_{cq} \in \mathbb{R}^{N_q \times C}$ that enables feature interaction among all auxiliary modalities. Specifically, the cross-modal queries interact with all the output tokens of uni-modal Q-formers $\{\hat{X}^m | m \in \mathcal{M}'\}$ with a cross-attention layer to capture cross-modality correlations among different auxiliary modalities, resulting in the cross-modality auxiliary features denoted as $\hat{X}^c$:

$$\hat{X}^c = \text{CrossAttn}\big(Q = X_{cq}, K = Z^a, V = Z^a\big), \tag{6}$$

where

$$Z^a = \text{Concat}(\{\hat{X}^m\}_{m \in \mathcal{M}'}). \tag{7}$$

Table 1: Results of three AD prediction tasks across three ADNI cohorts. Experiments are conducted under the Modality-complete setting, which includes only individuals with all four data modalities available. (**C**: Clinical records, **F**: fMRI, **S**: sMRI, **G**:Genetic data) The best results are in **bold**.

| Modality | Method | NC vs. MCI | | | NC vs. AD | | | sMCI vs. pMCI | | |
|---|---|---|---|---|---|---|---|---|---|---|
| | | ACC | SPE | SEN | ACC | SPE | SEN | ACC | SPE | SEN |
| *Uni-Modality* | | | | | | | | | | |
| C | RandomForest | 0.709 | 0.724 | 0.612 | 0.745 | 0.738 | 0.557 | 0.696 | 0.736 | 0.602 |
| C | LLaMA 2 | 0.793 | 0.854 | 0.640 | 0.814 | 0.879 | 0.687 | 0.721 | 0.809 | 0.574 |
| F | Brain-JePA | 0.777 | 0.838 | 0.542 | 0.807 | 0.857 | 0.576 | 0.714 | 0.723 | 0.522 |
| F | BrainLM | 0.768 | 0.809 | 0.537 | 0.781 | 0.841 | 0.575 | 0.705 | 0.735 | 0.509 |
| S | BrainMVP | 0.724 | 0.819 | 0.589 | 0.730 | 0.832 | 0.669 | 0.703 | 0.774 | 0.601 |
| S | SamMed3D | 0.714 | 0.807 | 0.597 | 0.714 | 0.814 | 0.675 | 0.689 | 0.758 | 0.607 |
| S | Swin-UNETR | 0.609 | 0.628 | 0.495 | 0.612 | 0.724 | 0.579 | 0.521 | 0.595 | 0.503 |
| S | M$^3$AE | 0.647 | 0.665 | 0.538 | 0.671 | 0.778 | 0.609 | 0.622 | 0.666 | 0.591 |
| G | NT-Human | 0.694 | 0.775 | 0.521 | 0.751 | 0.857 | 0.492 | 0.652 | 0.719 | 0.424 |
| G | SEI | 0.483 | 0.500 | 0.462 | 0.568 | 0.680 | 0.491 | 0.415 | 0.657 | 0.342 |
| G | DNA-Bert2 | 0.709 | 0.724 | 0.612 | 0.746 | 0.840 | 0.557 | 0.659 | 0.813 | 0.460 |
| *Multi-Modality* | | | | | | | | | | |
| C+G+F+S | M4Survive | 0.827 | 0.865 | 0.568 | 0.804 | 0.879 | 0.657 | 0.746 | 0.840 | 0.557 |
| C+G+F+S | LateFusion | 0.818 | 0.894 | 0.433 | 0.798 | 0.867 | 0.581 | 0.714 | 0.782 | 0.582 |
| **C+G+F+S** | **Ours** | **0.871** | **0.921** | **0.700** | **0.846** | **0.902** | **0.707** | **0.763** | **0.876** | **0.617** |

Table 2: Results of three AD prediction tasks under the Modality-incomplete setting. **\*** denotes the use of pretrained weights from the original paper for evaluation.

| Modality | Method | NC vs. MCI | | | NC vs. AD | | | sMCI vs. pMCI | | |
|---|---|---|---|---|---|---|---|---|---|---|
| | | ACC | SPE | SEN | ACC | SPE | SEN | ACC | SPE | SEN |
| *bi-Modality* | | | | | | | | | | |
| S+C | Ncomms | 0.945 | 0.932 | 0.947 | 0.928 | 0.939 | 0.911 | 0.773 | 0.766 | 0.698 |
| S+C | AI-diagnosis | 0.924 | 0.947 | 0.938 | 0.910 | 0.920 | 0.890 | 0.766 | 0.839 | 0.574 |
| S+C | AI-diagnosis* | 0.950 | 0.937 | 0.955 | 0.924 | 0.939 | 0.895 | 0.825 | 0.849 | **0.740** |
| S+C | SMART | 0.932 | 0.943 | 0.877 | 0.917 | 0.944 | 0.891 | 0.810 | 0.832 | 0.768 |
| *Multi-Modality* | | | | | | | | | | |
| C+G+F+S | M4Survive | 0.926 | 0.921 | 0.931 | 0.911 | 0.936 | 0.851 | 0.812 | 0.879 | 0.652 |
| C+G+F+S | LateFusion | 0.881 | 0.927 | 0.861 | 0.899 | 0.912 | 0.879 | 0.801 | 0.871 | 0.693 |
| **C+G+F+S** | **Ours** | **0.979** | **0.957** | **0.963** | **0.945** | **0.960** | **0.931** | **0.846** | **0.901** | 0.711 |

Finally, the cross-modal auxiliary feature $\hat{X}^c$ and a set of uni-modal auxiliary features $\{\hat{X}^m | m \in \mathcal{M}'\}$ are concatenated to obtain the final output of the modality-aware Q-former:

$$H^a = \text{Concat}\big(\{\hat{X}^m\}_{m \in \mathcal{M}'}, \hat{X}^c\big) \in \mathbb{R}^{4N_q \times C}. \tag{8}$$

# 4 EXPERIMENTS

## 4.1 DATASETS

We leverage the ADNI (Mueller et al., 2005) dataset to evaluate our method in AD diagnosis and progression prediction task. Alzheimer's Disease Neuroimaging Initiative (ADNI) offers the most comprehensive set of modalities, including structural and functional MRI (sMRI and fMRI), genetic data, and textual and tabular clinical records. ADNI includes participants across three main diagnostic categories: normal controls (NC), mild cognitive impairment (MCI), and Alzheimer's Disease (AD).

We use two datasets for external evaluation: PPMI (Marek et al., 2011) dataset focuses on Parkinson's disease, providing the same set of modalities as ADNI. It includes subjects across three diagnostic categories: normal controls (NC), mild cognitive impairment (MCI), and Parkinson's disease (PD). OASIS-3 (LaMontagne et al., 2019) is a multi-modal dataset providing sMRI, fMRI, and clinical records, but unlike ADNI, it lacks genetic data. It comprises subjects diagnosed as normal controls (NC) and Alzheimer's disease (AD). Further details on the three datasets and preprocessing pipelines for the four modalities are provided in the Appendix A.

Table 3: Results of PD prediction tasks on PPMI.

| Modality | Method | NC vs. MCI | | NC vs. PD | | pPD vs sPD | |
|---|---|---|---|---|---|---|---|
| | | ACC | AUC | ACC | AUC | ACC | AUC |
| C | LLaMA 2 | 0.857 | 0.839 | 0.913 | 0.909 | 0.694 | 0.681 |
| F | Brain-JePA | 0.808 | 0.797 | 0.871 | 0.875 | 0.652 | 0.669 |
| S | BrainMVP | 0.781 | 0.783 | 0.898 | 0.882 | 0.647 | 0.650 |
| G | NT-Human | 0.633 | 0.629 | 0.745 | 0.750 | 0.615 | 0.603 |
| S+C | Ncomms | 0.867 | 0.854 | 0.919 | 0.922 | 0.707 | 0.711 |
| S+C | SMART | 0.892 | 0.888 | 0.940 | 0.943 | 0.748 | 0.751 |
| S+C | AI-diagnosis | 0.909 | 0.905 | 0.934 | 0.948 | 0.731 | 0.739 |
| C+G+F+S | M4Survive | 0.889 | 0.910 | 0.954 | 0.968 | 0.752 | 0.753 |
| C+G+F+S | LateFusion | 0.860 | 0.846 | 0.940 | 0.951 | 0.707 | 0.719 |
| **C+G+F+S** | **Ours** | **0.927** | **0.944** | **0.967** | **0.969** | **0.769** | **0.773** |

Table 4: AD prediction task on OASIS.

| Modality | Method | NC vs. AD | |
|---|---|---|---|
| | | ACC | AUC |
| C | LLaMA 2 | 0.697 | 0.650 |
| F | Brain-JePA | 0.681 | 0.667 |
| S | BrainMVP | 0.655 | 0.621 |
| G | NT-Human | 0.491 | 0.489 |
| S+C | Ncomms | 0.662 | 0.636 |
| S+C | SMART | 0.701 | 0.679 |
| S+C | AI-diagnosis | 0.705 | 0.688 |
| C+G+F+S | M4Survive | **0.722** | 0.640 |
| C+G+F+S | LateFusion | 0.694 | 0.648 |
| **C+G+F+S** | **Ours** | **0.722** | **0.699** |

## 4.2 EXPERIMENTAL SETTINGS

In this study, we focus on two types of multi-modal diagnostic tasks: Alzheimer's disease (AD) diagnosis and prediction of prodromal progression, evaluated under both modality-complete and modality-incomplete settings.

**AD prediction.** For AD diagnosis evaluation, we follow established practices (Ning et al., 2021; Song et al., 2022; Hett et al., 2021; Feng et al., 2023a) by assessing performance on two binary classification tasks: normal controls (NC) vs. Alzheimer's Disease (AD) and NC vs. mild cognitive impairment (MCI). All models are evaluated under identical conditions to ensure a fair comparison using standard performance metrics: accuracy (ACC), specificity (SPE), and sensitivity (SEN).

**Prodromal progression prediction.** We extend the AD diagnosis task into a more challenging task of predicting AD progression (Rahim et al., 2023; El-Sappagh et al., 2020) by distinguishing between stable MCI (sMCI) and progressive MCI (pMCI). Both sMCI and pMCI patients are initially diagnosed with MCI at baseline, but the cognitive condition of sMCI group remained stable and did not convert to AD within 36 months after their first visit. In contrast, the pMCI group progressed to a clinical diagnosis of AD during the same 36-month follow-up period. (Ning et al., 2021) This task demands subtle discrimination between individuals with different temporal outcomes.

**Modality setting.** To comprehensively assess the robustness of our framework, we evaluate its performance on all three tasks under two different modality settings: *Modality-complete setting* includes only individuals with all four data modalities available. Under this strict requirement, from the ADNI1, ADNI2, and ADNI3 cohorts, we obtained 414 CN, 68 AD, and 273 MCI samples for two AD prediction tasks, and 182 sMCI and 44 pMCI samples for the progression prediction task. *Modality-incomplete setting* includes individuals with at least one available modality, better reflecting real-world clinical scenarios and enabling fuller data utilization. We collected 898 CN, 416 AD, and 986 MCI samples for AD prediction, as well as 220 sMCI and 81 pMCI samples for sMCI vs. pMCI.

**Cross-domain generalization setting.** To assess generalization ability, we conduct out-of-distribution (OOD) evaluation under modality-complete setting. Our framework is trained on NC vs. AD data from ADNI. For OOD testing, we use 120 NC and 42 AD subjects from OASIS-3, all of which are unseen during training and with all three modalities (sMRI, fMRI, and clinical records) available. Predictions are obtained by combining the outputs of the three corresponding foundation models trained on ADNI. To further demonstrate the adaptability of our method to other brain diseases, we train our framework on PPMI dataset and evaluate on *Parkinson's disease (PD) diagnosis* and *progression prediction*. Similar to AD, PD prediction assesses the performance on NC vs. PD and NC vs. MCI tasks, and prodromal progression prediction of PD follows the same protocol to distinguish between stable PD (sPD) and progressive PD (pPD). Under modality-complete setting, we obtained 743 NC, 329 PD, and 143 MCI samples for NC vs. PD and NC vs. MCI classification, as well as 225 sPD and 104 pPD samples for sPD vs. pPD classification.

**Implementation Details.** In our framework, each modality is processed by a dedicated pre-trained foundation model: LLaMA2-13B (Touvron et al., 2023) for clinical records, Brain-JEPA (Dong et al., 2024) for fMRI, BrainMVP (Rui et al., 2025) for sMRI, and NT-500M (Dalla-Torre et al., 2025) for genetic data. Full details of models, data split, and hyper parameters are in the Appendix B.

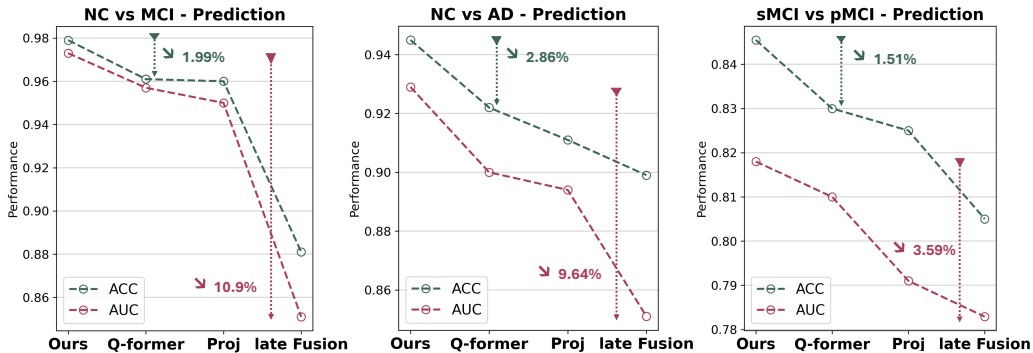

Figure 2: Analysis of Anchor and Auxiliary modality fusion methods for AD Diagnosis on ADNI

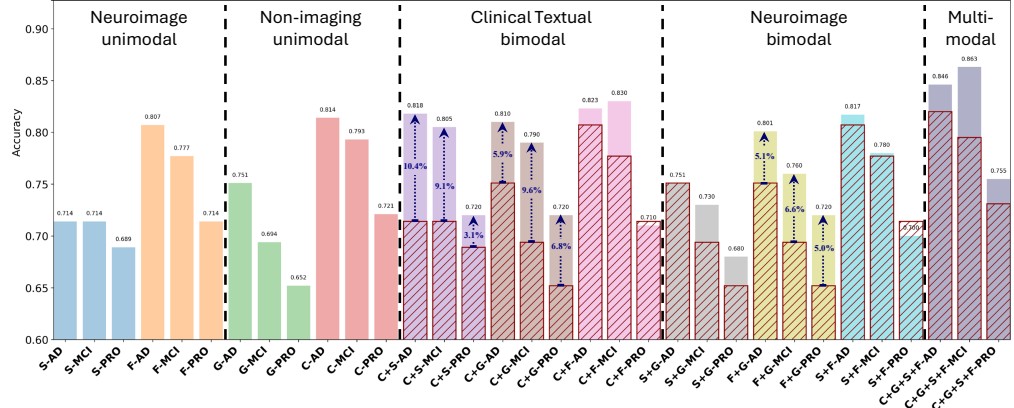

Figure 3: Overall performance trend with different modality combinations. **C**: Clinical records; **F**: fMRI; **S**: sMRI; **G**: Genetic data. Red striped bars indicate performance where one modality is removed from a given multi-modal combination. Figure Details are in the Appendix D.7.

## 4.3 MAIN RESULTS

*Baselines.* We compare our proposed method against representative uni-modality (Rigatti, 2017; Touvron et al., 2023; Dong et al., 2024; Caro et al., 2024; Rui et al., 2025; Wang et al., 2023; Tang et al., 2022; Liu et al., 2023b; Nguyen et al., 2023; Chen et al., 2022; Zhou et al., 2024), bi-modality (Xue et al., 2024; Chen & Hong, 2024; Qiu et al., 2022), and multi-modality models (Lee et al., 2025) on diagnosis and progression prediction tasks. Baselines' details are in the Appendix D.1.

*AD classification Results.* We compare our method with SOTA baselines under both modality-complete and incomplete settings on ADNI, with results presented in Table 1 and 2. Overall, our method outperforms all baselines in overall accuracy across all three AD classification tasks under two settings. Compared to conventional Multi-Modal methods including M4Survive and Late Fusion, our Modality-anchored Interaction ensures the anchor modality feature space to be preserved while integrating sufficient auxiliary features into the anchor representation. Therefore, our joint interaction and adoption framework is robust to both limited data availability and missing modalities. Modality-anchored interaction between anchor and auxiliary modalities preserves well-structured feature space of each pre-trained model and avoids representation degradation caused by missing modality inputs. For each modality, we select the most recent and representative foundation models and evaluate their uni-modal performance on the ADNI dataset in Table 1. Among the baselines, LLaMA2 (clinical records), BrainMVP (sMRI), BrainJePA (fMRI), and NT-Human (genetic data) achieve the highest overall performance for their respective modalities.

*Cross-Domain Generalization.* Table 3 demonstrates that our method generalizes effectively beyond AD, significantly outperforming both uni- and multi-modal baselines in Parkinson's disease diagnosis on the PPMI dataset. Similarly, Table 4 shows that when transferring a model trained on ADNI to the out-of-distribution dataset like OASIS-3, our approach achieves state-of-the-art AUC performance.

Table 5: Fusion baselines comparison under modality-incomplete setting on ADNI.

| Methods | NC vs MCI | | NC vs AD | |
|---|---|---|---|---|
| *Fusion Baselines* | ACC | AUC | ACC | AUC |
| Feature Concatenation | 0.894 | 0.885 | 0.833 | 0.846 |
| Linear Fusion | 0.881 | 0.851 | 0.899 | 0.851 |
| Self-Attention Fusion | 0.921 | 0.917 | 0.901 | 0.861 |
| *Modality-anchored interaction (Ours)* | | | | |
| - Train from pre-trained | **0.979** | **0.969** | **0.945** | **0.944** |

Table 6: Ablation study on choice of foundation models for each modality on ADNI.

| Modality | Foundation Models | NC vs. MCI | NC vs.AD | sMCI vs. pMCI |
|---|---|---|---|---|
| sMRI | Sammed 3D (Wang et al., 2023) | 0.961 | 0.941 | 0.825 |
| | vs. BrainMVP | -0.018 | -0.004 | -0.021 |
| Genetic | DNA-Bert 2 (Zhou et al., 2024) | 0.960 | 0.931 | 0.830 |
| | vs. NT-Transformer | -0.019 | -0.013 | -0.016 |
| fMRI | BrainLM (Caro et al., 2024) | 0.958 | 0.924 | 0.839 |
| | vs. BrainJePA | -0.021 | -0.021 | -0.007 |
| Clinical records | MedGemma (Sellergren et al., 2025) | 0.967 | 0.931 | 0.813 |
| | vs. Llama 2 | -0.012 | -0.014 | -0.033 |

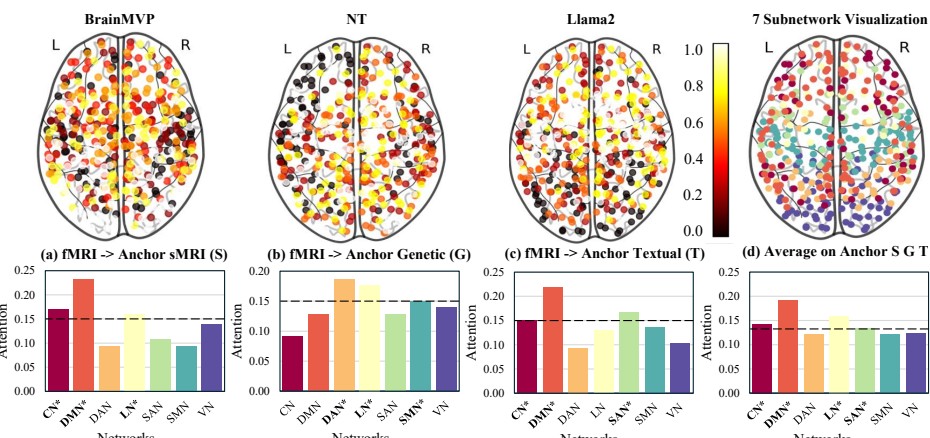

Figure 4: Attention maps of fMRI images across three anchor modalities and their models. Attention across seven brain networks is computed for NC/MCI classification.

These results highlight both the adaptability of our framework across different neurodegenerative diseases and its strong generalization ability to unseen AD datasets.

## 4.4 ABLATION STUDY

**Effectiveness of key components.** To assess the effectiveness of our proposed modality-aware Q-former, we replace it with two alternative fusion strategies in Fig. 2: (1) Linear Projection (Proj), where features from each modality are projected to a shared space and concatenated; (2) Q-Former, where features from each modality interact only with their own learnable query tokens via attention and are then concatenated. While both Proj and Q-Former surpass the late-fusion baseline, their ability to align heterogeneous modalities is limited. Our proposed modality-aware Q-Former consistently outperforms those baselines, demonstrating the effectiveness of our cross-modal query design in extracting synergistic information while preserving the structured representations of each modality.

**Comparison with fusion baselines.** In Table 5, we compare our modality-anchored interaction with three fusion baselines using the same pre-trained foundation models. Feature Concatenation: concatenate features from all modalities. Linear Fusion / Self-Attention: project features from each modality into a shared space using linear / self-attention layer. Our input-level interaction method achieves superior performance and shows richer modality integration than output-level fusion.

**Performance trend with different modality combinations.** Fig 3 presents the performance trend from uni-modal to bi-modal and full multi-modal. Combining modalities generally improves performance (denoted by red striped bars), with clinical records and fMRI showing the most significant performance gains. Using all four modalities (C+G+S+F) achieves the highest performance, showing the complementary nature of cross-modal information and the effectiveness of our method to enable sufficient interaction among foundation models.

**Number of queries.** Fig 5 shows an ablation study on the number of queries. When set to 0, the model degrades to a late-fusion baseline. As the number increases, the ACC performance on two tasks

improves on the ADNI dataset. The performance with 16 queries suggests that sufficient cross-modal interaction has been achieved at this point.

**Ablation on foundation model selection.** To validate our foundation model choices, we conduct ablation studies by replacing the foundation model of one modality at a time while keeping the others fixed. Accuracy comparisons in Table 6 show that our selected models consistently achieve better performance, confirming their suitability for our framework.

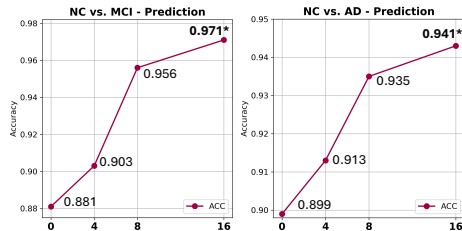

Figure 5: Query numbers ablation.

**Visualization and Interpretability.** We visualize the attention maps of fMRI images in the NC/MCI classification task. In Stage 2, each anchor modality model takes fMRI representations as input. To identify which fMRI regions are most attended to, we compute the average attention weights that anchor models' embeddings assign to each brain parcel using the Attention-Rollout method (Abnar & Zuidema, 2020). With the Schaefer functional atlas (Schaefer et al., 2018), the brain network is partitioned into seven subnetworks. We then average the ROI attention values within each subnetwork and normalize them to derive the network-level attention distribution. As shown in Fig. 4, when serving as the anchor modality, sMRI predominantly attends to the DMN, LN, and CN; genetic data emphasizes the DAN, LN, and SMN; and clinical records highlight the DMN, SAN, and CN. Our findings show that modality-anchored interaction enables each anchor modality to selectively focus on its most discriminative fMRI regions while facilitating complementary information exchange across modalities. Averaged across the three anchor modalities (Fig. 4 d), the attention emphasizes the roles of DMN, CN, SAN, and LN in cognitive impairment, in line with prior research (Talwar et al., 2021; Sheline & Raichle, 2013; Brier et al., 2014). (More visualization in Appendix D.5)

## 5    CONCLUSION

In this paper, we propose a unified multi-modal framework for Alzheimer's Disease diagnosis that leverages pre-trained uni-modal models with modality-anchored interaction and modality-aware Q-Former to enable early and effective anchor and auxiliary modalities interaction. Broad evaluations on ADNI, PPMI, and OASIS-3 demonstrate the strong performance and generalization of our framework, underscoring the potential of foundation model adaptation for medical multi-modal learning.

## 6    ACKNOWLEDGMENT

This work was supported by National Natural Science Fund of China (No. U25A20527, 62473286). This work was also supported by Shanghai Municipal Science and Technology Major Project (No. 2025SHZDZX025G10).

## REPRODUCIBILITY STATEMENT

To support replication, we provide implementation details of our method in Appendix B, including settings, data splits, and training hyper parameters for experiments and foundation models across four modalities. The training parameters of modality-aware Q-former are in Appendix D.8. The implementation details of compared methods are in Appendix D.1. The preprocessing pipelines for each modality are described in Appendix A. Together, these details reflect practical considerations for real-world deployment.

## ETHICS STATEMENT

This work adheres to the ICLR Code of Ethics. No human or animal experiments were conducted. All datasets were publicly available and used in accordance with their respective guidelines, ensuring privacy protection. We are committed to maintaining ethical standards and fostering responsible AI use.

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

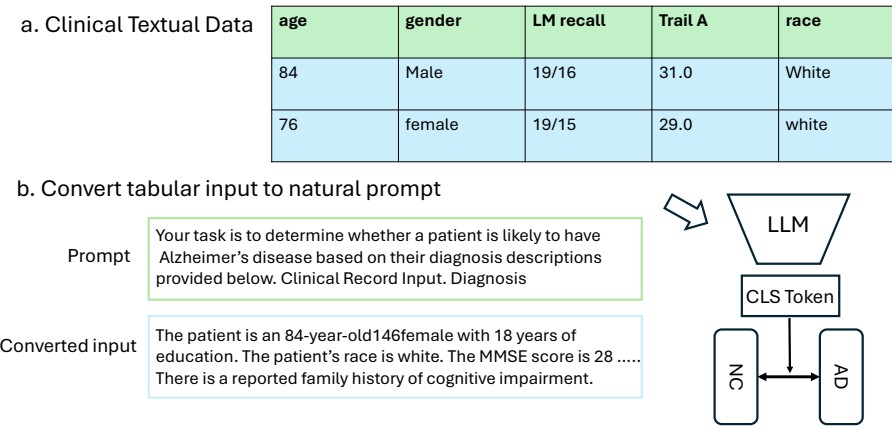

Figure 6: Tabular Data Prompt Construction Pipeline.

## A  DATA CONSOLIDATION AND AUGMENTATION

### A.1  CLINICAL RECORDS PROMPT TEMPLATE.

To use a LLM for clinical tabular records, the tabular input and classification task must be transformed into natural text. Fig 6 give a instance of tabular data.

**Tabular Data.** Our tabular data includes a set of clinical features covering multiple main categories: demographic information (e.g., age, gender, education, ethnicity, and race), genetic risk factors (APOE genotype), cognitive assessment scores (including MMSE, CDR, MoCA, and other neuropsychological tests), neuropsychiatric symptoms (from item-level NPI-Q responses), functional ability metrics (from the Functional Activities Questionnaire), medical history and lifestyle indicators (e.g., cardiovascular conditions, psychiatric disorders, smoking and alcohol use), and imaging-related parameters such as scanner strength.

**Task prompt** To serialize structured tabular data into natural language prompts, we adopt a text templating strategy that systematically enumerates each feature along with its corresponding value in natural language form. The detailed prompt and prefix are as follows:

**Listing 1: Task Prompt.**

```
Task prompt (AD vs. NC) = '''
Your task is to determine whether a patient is likely to have an
    Alzheimer's disease based on their diagnosis descriptions
    provided below.
'''
Task prompt (MCI vs. NC) = '''
Your task is to determine whether a patient is likely to have mild
     cognitive impairment based on their diagnosis descriptions
    provided below.
'''
Task prompt (sMCI vs. pMCI) = '''
Your task is to determine whether a patient with mild cognitive
    impairment is likely to remain stable or progress to Alzheimer'
    s disease based on their diagnosis descriptions provided below.

'''
```

**Description Prompt.**   To generate the description prompt, for categorical and numerical attributes, we combine the name and its values into a sentence. And for binary features, we include the feature name in the prompt only if the value is True to avoid generating unnecessary and false information.

**Listing 2: Description Prompt.**

```
Description Prompt = '''
The patient is a {age}-year-old {gender} with {education} years of
    education. Their ethnicity is coded as {hispanic}, and their
    race is {race}. The APOE status is {apoe}. The MMSE score is {
    mmse}, and the CDR score is {cdr} with a sum of boxes score of
    {cdrSum}. MRI Tesla strength is recorded as {Tesla}.
    Cognitive test results include Trail A ({trailA}), Trail B ({
    trailB}), LM immediate recall ({lm_imm}), LM delayed recall ({
    lm_del}), Boston naming test ({boston}), Animal fluency ({
    animal}), Digit span backward ({digitB}), Digit Span Backward
    Longest ({digitBL}), Digit span forward ({digitF}), Digit Span
    Forward Longest ({digitFL}). Neuropsychiatric inventory (NPI-
    Q) results indicate symptoms of {Mild/Moderate/Severe +
    symptom list}, or: Neuropsychiatric inventory (NPI-Q) results
    indicate no reported symptoms. Functional assessment (FAQ)
    shows difficulties in {faq_BILLS (label), faq_TAXES (label),
    ..., faq_TRAVEL (label)}. Medical history includes {
    Cardiovascular events (Yes), Psychiatric disorders (Yes), ...,
    Other depression-related conditions (Yes)}. GDS score: {gds}.
    MoCA score: {moca}.
'''
```

## A.2   GENOMIC DATASETS CONSTRUCTION.

**Construction**   Fig 7 illustrates the genomic datasets construction process. Firstly, we construct a genetic dataset with whole genome sequencing (WGS) studies for Alzheimer's disease (AD) Mueller et al. (2005), which provides base-pair level coverage of the entire genome, allowing for a comprehensive assessment of individual genetic variation. To enable downstream machine learning models to process genetic data within their input length limitations, for ADNI, we select two major genes, APOE and TOMM40, as the primary sources of genetic input. For PPMI, we select RIMS2 and TMEM108. These genes have been identified as being most strongly associated with AD or PD susceptibility Saykin et al. (2010). For each individual, up to 80 genetic variants were selected from two major genomic regions. Then a sequence extractor is used to convert those genetic variants records by locating the position of variants in the human reference genome.

**Usage**   Following Li et al. (2024b), for each variant, we located its position in the GRCh38/hg38 human reference genome and reconstructed the corresponding reference (ref) and alternate (alt) sequences. Variants located on rare configs not present in the reference genome were filtered out. These sequences were then concatenated separately, namely refs with refs and alts with alts, to construct a pair of representation of ref and alt sequences per individual, capturing AD-relevant genomic features for downstream prediction tasks.

**Genetic Variants Record Formulation**   Following this construction process in Fig 7, the minimum unit of ADNI gene dataset is individual variants record, which is an $(x, y)$ pair. Here, $x = (ref, alt)$, and $y$ denotes the individual's diagnostic label.

## A.3   MRI PRE-PROCESS PIPELINE.

We adopted two distinct pre-processing pipelines for structural and functional MRI data respectively.

**The pre-processing pipeline for sMRI.**   We use the T1-weighted MRI scan closest to each subject's baseline visit. To ensure consistency and quality, we exclusively selected 3T MRI scans, as they

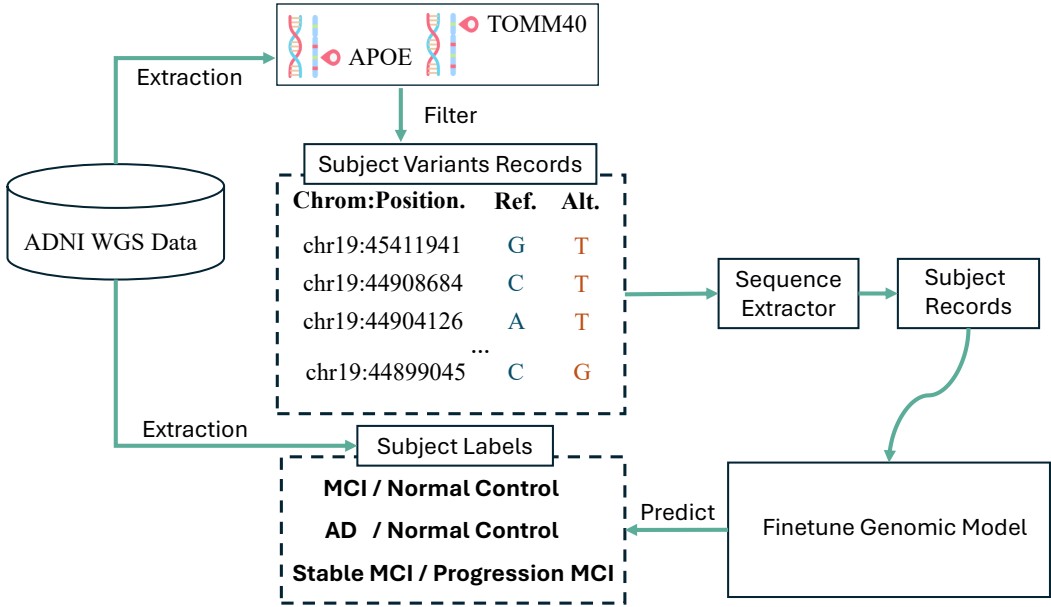

Figure 7: **Genetic Dataset Construction and Usage.** The workflow of gene dataset from ADNI dataset. Genetic variant records are first extracted from selected APOE and TOMM40 gene for each individual. These records are paired with diagnostic labels of individual derived from the ADNI dataset. A sequence extractor then organizes the variants into fixed-length sequences, which serve as model inputs for downstream genetic foundation models.

Table 7: Multimodal modalities and subject distributions across datasets.

| Datasets | Type | Modalities | #NC | #MCI | #AD | #PD |
|---|---|---|---|---|---|---|
| ADNI | NC/MCI/AD | C+G+F+S | 898 | 986 | 416 | - |
| OASIS | NC/AD | C+F+S | 120 | - | 43 | - |
| PPMI | NC/MCI/PD | C+G+F+S | 743 | 143 | - | 329 |

provide higher signal-to-noise ratio, better spatial resolution, and improved image quality compared to 1.5T scans, which benefits downstream learning. Pre-processing structural MRI specifically involves resampling all 3D structural T1-weighted (T1w) MRI volumes to a standardized spatial resolution of $128 \times 128 \times 128$, in accordance with the input requirements of the SAM-Med3D Wang et al. (2023) pretrained model.

**The pre-processing pipeline for fMRI.** Following the preprocessing pipeline of Brain-JEPA Dong et al. (2024), functional MRI data were preprocessed using the fMRIPrep pipeline Esteban et al. (2019), which incorporates high-precision anatomical reference via T1w MRI images. The pipeline included skull stripping, cortical surface reconstruction, slice-timing correction, followed by co-registration of fMRI time series to the corresponding anatomical scans. Spatial normalization to the MNI152 standard space was subsequently performed using nonlinear transformation. Following normalization, the fMRI time series were parcellated into $n = 450$ regions of interest (ROIs), with cortical regions delineated by the Schaefer-400 atlas Schaefer et al. (2018) and subcortical regions defined using the Tian-Scale III atlas Tian et al. (2020). To enhance inter-subject comparability and mitigate inter-participant variability, robust scaling was applied to each ROI by subtracting the across-subject median and dividing by the interquartile range Caro et al. (2023).

### A.4 Datasets Details.

As illutstrated in Table 7, three public multi modal datasets are evaluated in our study, considering four medical modalities, including two imaging modalities, genetic data, and clinical records.

***ADNI.*** We evaluate our method using the Alzheimer's Disease Neuroimaging Initiative (ADNI) dataset Mueller et al. (2005), which is the publicly available dataset that offers the most comprehensive set of modalities, including structural and functional MRI (sMRI and fMRI), genetic data, and textual and tabular clinical records. ADNI includes participants across three main diagnostic categories: normal controls (NC), mild cognitive impairment (MCI), and Alzheimer's Disease (AD).

***PPMI.*** PPMI is a public dataset Marek et al. (2011) focused on Parkinson's disease, providing the same set of modalities as ADNI. It includes subjects across three diagnostic categories: normal controls (NC), mild cognitive impairment (MCI), and Parkinson's disease (PD).

***OASIS-3.*** The OASIS-3 dataset LaMontagne et al. (2019) is a multimodal neuroimaging and clinical record resource that provides sMRI, fMRI, and textual and tabular clinical information, but lacks the genetic data available in ADNI. It comprises subjects diagnosed as normal controls (NC) and Alzheimer's disease (AD).

## B Pretrained Feature Extraction Implementation Details

### B.1 Settings of Experiments.

All experiments were conducted on NVIDIA A100 80GB GPUs, with a total of 32 GPUs used. Each compute worker was equipped with 64 CPU cores and 512 GB of RAM. Training under the modality-complete setting typically took around 1.5 hours per epoch. We adopt 16 as the number of queries in our proposed Modality-aware Q-former. Both Stage 1 (uni-modal foundation model adaptation) and Stage 2 (Modality-anchored interaction), illustrated in Fig 1, are trained in a supervised setting. The ground truth labels are determined based on established clinical diagnostic criteria Mueller et al. (2005). For the three datasets, we performed strict de-duplication of subjects using their unique IDs across all phases to ensure that each subject appears only once in the entire dataset. This guarantees that there is no overlap of subject data between the training, validation, and test sets, effectively eliminating the risk of data leakage. The dataset is partitioned into 60% for training, 20% for validation, and 20% for testing. We conducted test experiments using 5-fold cross-validation. For each fold, we record the performance on the test set. This resulted in five paired performance values, one for each fold. We then conducted statistical significance test through a paired t-test on these fold-wise results to assess whether the performance difference was statistically significant. The resulting p-value below 0.01 (p = 0.005) confirms that our performance improvements reported in the main results over the second-best aproach have been confirmed to be statistically significant. Our approach treats each modality as the anchor one in turn. As a result, in modality-incomplete setting, when one modality is missing, the remaining modalities can still serve as anchor inputs, and their features are extracted using the corresponding foundation models without disruption. In practice, when a modality is absent, we simply omit passing data to its associated Q-Former. For the Q-Formers of the remaining modalities, we set the query tokens corresponding to the missing modality to zero and apply an attention mask to prevent the model from attending to it.

### B.2 Settings and Hyper Parameters of Foundation Models.

Each modality is processed by a dedicated pre-trained foundation model: LLaMA2-13B Touvron et al. (2023) for textual data, NT Dalla-Torre et al. (2025) for genetic data, Brain-JEPA Dong et al. (2024)for fMRI, and BrainMVP Rui et al. (2025) for sMRI. Training details of each model is presented below.

**Textual feature Extraction.** The LLaMA2 model Touvron et al. (2023) in Fig 8 (Left) was leveraged to extract textual features from clinical records into a latent space representation. LLaMA 2 is an auto-regressive language model based on an optimized transformer architecture. The fine-tuned variants are aligned with human preferences through a two-stage process: supervised fine-tuning (SFT) on curated instruction-following data, followed by reinforcement learning with human feedback (RLHF) to further optimize for helpfulness, factuality, and safety. We utilize LLaMA2-12b-HF pre-traind weights. For LLaMA2-13B, we use the AdamW optimizer with a learning rate of 2e-4. The

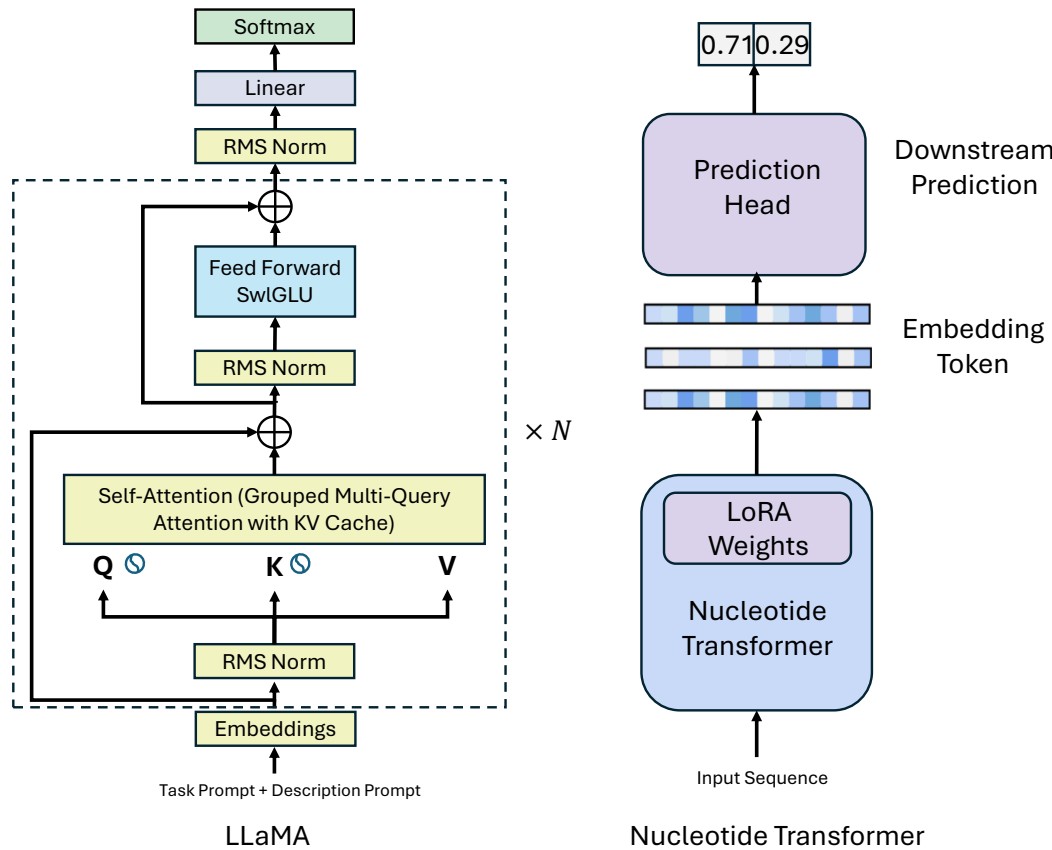

Figure 8: Architecture of LLaMA 2 (left) and NT-Human500 (right).

model is trained for a total of 6 epochs with a batch size of 8. We apply LoRA Hu et al. (2022) using the Hugging Face peft library with the following configuration: rank 32, $\alpha = 32$, $dropout = 0.1$, and no bias adaptation for classification.

**Genomic Feature Extraction.** The Nucleotide Transformer Dalla-Torre et al. (2025) in Fig 8 (right), was leveraged to extract genomic features from ADNI Gene Dataset constructed in Fig 7. The Nucleotide Transformers are foundational language models pre-trained on DNA sequences from whole genomes. Unlike traditional approaches that rely solely on a single reference genome, our models incorporate genetic information from over 3,200 diverse human genomes and 850 genomes across various species, including both model and non-model organisms. Through comprehensive evaluations, we demonstrate that these large-scale models significantly outperform existing methods in predicting molecular phenotypes with high accuracy. We utilize nucleotide-transformer-500m-human-ref pre-traind weights, a 500M parameters transformer pre-trained on the human reference genome. For NT-500M-Human, we use the AdamW optimizer with a learning rate of 5e-4. The model is trained for 2 epochs with a batch size of 4. We apply LoRA Hu et al. (2022) using the Hugging Face peft library with the following configuration: rank 32, $\alpha = 32$, $dropout = 0.1$, and no bias adaptation for classification.

**Neuro-imaging Feature Extraction** BrainJEPA Dong et al. (2024) was employed to extract latent representations from fMRI scans, leveraging pretraining on large-scale 3D brain MRI data. We adopt jepa-ep300 weights, the pre-trained model weights resulting from training on the UKB dataset. The target encoder from this model serves as the feature extractor for fMRI data. We also adopted BrainMVP Rui et al. (2025) for sMRI representation. Its fully 3D architecture includes a BrainMVP image encoder and BrainMVP decoder, built with UniFormer backbone to capture spatial features

Table 8: Effect of Connectors. The ACC and AUC performance across three AD diagnosis and progression prediction tasks on ADNI dataset in modality-incomplete settings is reported.

| Components | NC vs. MCI | | NC vs. AD | | sMCI vs. pMCI | |
|---|---|---|---|---|---|---|
| | ACC | AUC | ACC | AUC | ACC | AUC |
| late Fusion | 0.881 | 0.851 | 0.899 | 0.851 | 0.805 | 0.783 |
| Proj | 0.961 | 0.957 | 0.911 | 0.904 | 0.825 | 0.791 |
| Q-former | 0.960 | 0.960 | 0.915 | 0.910 | 0.830 | 0.810 |
| Modality-Aware Q-former | **0.979** | **0.969** | **0.945** | **0.944** | **0.846** | **0.818** |

effectively. We utilize 16k mpMRI weights. For Brain-JEPA, training is conducted for 200 epochs with a batch size of 8, using AdamW with an initial learning rate of 5e-5. For BrainMVP, we apply AdamW with learning rate of 8e-4 and train the model for 200 epochs with a batch size of 4.

## C  IMPLEMENTATION DETAILS OF Q-FORMER.

### C.1  DETAILS OF MODALITY-AWARE Q-FORMERS

To enable interaction between the representations from the anchor and auxiliary modalities, we propose to employ a set of learnable queries to explicitly project the auxiliary features into the representation space of the anchor one. As shown in Eq 2, to allow effective interaction between the anchor model and the auxiliary models, a transformer-based connector is proposed to selectively project features from the auxiliary modality to the feature space of the anchor model, called modality-aware Q-former. As illustrated in Fig 1, our modality-aware Q-former incorporates two types of information, namely uni-modal and cross modal information. Specifically, we create a set of learnable queries $X \in \mathbb{R}^{N_q \times C}$ $x \in \mathbb{R}^{C}$ with number of learnable query tokens $N_q$, serving as the query:

$$X \in \mathbb{R}^{N_q \times C}, \tag{9}$$

**Number of learnable query tokens.** We set the number of learnable query tokens $N_q = 16$ to strike a balance between leveraging auxiliary modality information and maintaining stable training of the anchor modality. As illustrated in Fig 5, increasing $N_q$ allows the model to incorporate more information from auxiliary modalities, enhancing cross-modal interactions and semantic representation. Conversely, reducing $N_q$ limits the contribution of auxiliary modalities; when $N_q$ approaches zero, the model effectively degrades to a late fusion strategy, where only uni-modal representations from the anchor modality are used without cross-modal guidance.

**Uni-modal Q-formers** Modality-aware Q-former first extracts the uni-modality information from a specific auxiliary modality $m \in \mathcal{M}'$. Specifically, we create a set of learnable tokens to serve as uni-modality queries, denoted as $X_{uq} \in \mathbb{R}^{N_q \times C}$. Given the auxiliary features extracted from the corresponding auxiliary model $F_m$, we first project them to the same dimension as the anchor modality:

$$Z^m = \text{Linear}\big(F_m(X^m)\big) \in \mathbb{R}^{L^m \times C}. \tag{10}$$

Then, the learnable uni-modal queries interact with the projected features through a cross-attention layer, which further projects the auxiliary modality features into the anchor modality feature space and extracts information relevant to the anchor modality from the auxiliary one $m$:

$$\hat{X}^m = \text{CrossAttn}(Q = X_{uq}^m, K = Z^m, V = Z^m). \tag{11}$$

The resulting output $\hat{X}^m \in \mathbb{R}^{N_q \times C}$ are features containing uni-modal information from auxiliary modality $m$.

**Cross-modal Q-former** Besides uni-modal information, we further propose a set of cross-modal queries $X_{cq} \in \mathbb{R}^{N_q \times C}$ that enables feature interaction among all auxiliary modalities. Specifically, the cross-modal queries interact with all the output tokens of uni-modal Q-formers $\{\hat{X}^m | m \in \mathcal{M}'\}$ with

a cross-attention layer to capture cross-modality correlations among different auxiliary modalities, resulting in the cross-modality auxiliary features denoted as $\hat{X}^c$:

$$\hat{X}^c = \text{CrossAttn}\big(Q = X_{cq}, K = Z^a, V = Z^a\big), \tag{12}$$

where

$$Z^a = \text{Concat}(\{\hat{X}^m\}_{m \in \mathcal{M}'}). \tag{13}$$

Finally, the cross-modal auxiliary feature $\hat{X}^c$ and a set of uni-modal auxiliary features $\{\hat{X}^m | m \in \mathcal{M}'\}$ are concatenated to obtain the final output of the modality-aware Q-former:

$$H^a = \text{Concat}\big(\{\hat{X}^m\}_{m \in \mathcal{M}'}, \hat{X}^c\big) \in \mathbb{R}^{2N_q \times C}. \tag{14}$$

**Q-former and Linear dimensions**  For anchor modality $m \in \mathcal{M} = \mathtt{s}, \mathtt{f}, \mathtt{c}, \mathtt{g}$ and its uni-modal learnable query tokens $X_{uq}^m \in \mathbb{R}^{N_q \times C_m}$ and cross-modal query tokens $X_{cq} \in \mathbb{R}^{N_q \times C}$, the dimensionality $C_m$ for each modality is defined as follows:

$$C_m = \begin{cases} 512, & \text{if } m = \mathtt{s} \\ 768, & \text{if } m = \mathtt{f} \\ 5120, & \text{if } m = \mathtt{c} \\ 1280, & \text{if } m = \mathtt{g}. \end{cases} \tag{15}$$

### C.2 EFFECTIVENESS OF CROSS-MODALITY Q-FORMER.

Compared with conventional fusion methods such as Late-Fusion, which combine uni-modal predictions at the output level, our Q-Former design enables earlier and more effective cross-modal interactions at the level of anchor token embeddings and auxiliary features, allowing more effective integration of complementary information from all modalities. To isolate the effect of modality interaction mechanisms and assess the effectiveness of our proposed Modality-aware Q-former, we consider a setting where all four modalities (C/F/S/G) are available, and apply four types of interaction strategies on top of pretrained uni-modal foundation models from stage 1. Specifically, we apply four types of fusion strategies: (1) Linear Projection (Proj), where features from each modality are projected to a shared space with simple linear modules and concatenated; (2) Late-Fusion, where each modality is independently processed and final predictions are aggregated; (3) Q-Former, where concatenated auxiliary modality embeddings serve as keys and values in the cross-attention layer, and learnable query tokens attend to them, enabling interaction with the anchor modality. (3) Modality-aware Q-Former, which introduces attention-based query tokens to adaptively gather relevant features from auxiliary modality.

For fair comparisons, we freeze the uni-modal pretrained models from stage 1 and only train the fusion modules and anchor models. Results are reported in Table 8 under Modality-Incomplete scenarios. While both Projection and Late-Fusion benefit from pre-trained features, their ability to align heterogeneous modalities is limited. Projection performs better than Late-Fusion on general classification accuracy. Our Q-Former consistently outperforms both baselines across all tasks. This demonstrates the effectiveness of our cross-modal query design in extracting synergistic information while preserving the structured representations of each modality.

## D  EXTENDED EXPERIMENTAL ANALYSIS

### D.1  IMPLEMENTATION DETAILS OF COMPARED BASELINES.

We compare our approach with SOTA multi-modal baselines. M4Survive Lee et al. (2025), Ncomms Qiu et al. (2022), Smart Chen & Hong (2024), and AIdiagnosis Xue et al. (2024) are SOTA multi-modality framework. M4Survive leverages various Radiology-Pathology pre-trained models to independently process and generate modality-specific embeddings. Then the embeddings are mapped into a joint-modality feature space and processed by a Mamba adapter to perform interaction for downstream prediction. M4Survive adopts LLaMA, BrainMVP, BrainJEPA, and

Table 9: Comparison to Uni-modal baselines. The performance across three AD diagnosis and progression prediction tasks in modality-complete settings is reported.

| Modality | Method | NC vs. MCI | | | NC vs. AD | | | sMCI vs. pMCI | | |
|---|---|---|---|---|---|---|---|---|---|---|
| | | ACC (%) | SPE (%) | SEN (%) | ACC (%) | SPE (%) | SEN (%) | ACC (%) | SPE (%) | SEN (%) |
| C | RandomForest Rigatti (2017) | 0.709 | 0.724 | 0.612 | 0.745 | 0.738 | 0.557 | 0.696 | 0.736 | 0.602 |
| C | LLaMA 2 Touvron et al. (2023) | 0.793 | 0.854 | 0.640 | 0.814 | 0.879 | 0.687 | 0.721 | 0.809 | 0.574 |
| F | Brain-JePA Dong et al. (2024) | 0.777 | 0.838 | 0.542 | 0.807 | 0.857 | 0.576 | 0.714 | 0.723 | 0.522 |
| F | BrainLM Caro et al. (2024) | 0.768 | 0.809 | 0.537 | 0.781 | 0.841 | 0.575 | 0.705 | 0.735 | 0.509 |
| S | BrainMVP Rui et al. (2025) | 0.724 | 0.819 | 0.589 | 0.730 | 0.832 | 0.669 | 0.703 | 0.730 | 0.640 |
| S | SamMed3D Wang et al. (2023) | 0.714 | 0.807 | 0.597 | 0.714 | 0.814 | 0.675 | 0.689 | 0.718 | 0.647 |
| S | Swin-UNETR Tang et al. (2022) | 0.609 | 0.628 | 0.495 | 0.612 | 0.724 | 0.579 | 0.521 | 0.595 | 0.503 |
| S | M$^3$AE Liu et al. (2023b) | 0.647 | 0.665 | 0.538 | 0.671 | 0.778 | 0.609 | 0.622 | 0.666 | 0.591 |
| G | NT-Human Nguyen et al. (2023) | 0.694 | 0.775 | 0.521 | 0.751 | 0.857 | 0.492 | 0.652 | 0.719 | 0.424 |
| G | SEI Chen et al. (2022) | 0.483 | 0.500 | 0.462 | 0.568 | 0.680 | 0.491 | 0.415 | 0.657 | 0.342 |
| G | DNA-Bert2 Zhou et al. (2024) | 0.709 | 0.724 | 0.612 | 0.746 | 0.840 | 0.557 | 0.659 | 0.813 | 0.460 |

Table 10: Results of three AD prediction tasks across the ADNI-1, ADNI-2, and ADNI-3 cohorts. Experiments are conducted under the Modality-Incomplete setting, indicating the presence of partial clinical textual records, functional MRI, structural MRI, and genetic data.

| Methods | NC vs. MCI | | NC vs. AD | | pMCI vs. sMCI | |
|---|---|---|---|---|---|---|
| | ACC | AUC | ACC | AUC | ACC | AUC |
| Feature Concatenation | 0.894 | 0.885 | 0.833 | 0.846 | 0.771 | 0.750 |
| Linear Classifier | 0.881 | 0.851 | 0.899 | 0.851 | 0.805 | 0.783 |
| Self-Attention Fusion | 0.921 | 0.917 | 0.901 | 0.861 | 0.785 | 0.751 |
| **Ours** | **0.961** | **0.969** | **0.945** | **0.944** | **0.825** | **0.846** |

NT as modality-specific encoders for text, sMRI, fMRI, and genomic modalities to adapt to our AD analysis tasks. After fine-tuning the pretrained encoders on our diagnosis tasks, We freeze the modality-specific foundation models and train the adapter network with a batch size of 16, a learning rate of 0.0003, over 30 epochs. Experiments employing MLP and transformer are executed on a NVIDIA A100 GPU.

Ncomms introduces a deep learning framework that comprises three components: (1) an MRI-only CNN model, (2) non-imaging models based on traditional machine learning classifiers, and (3) a hybrid fusion model that integrates imaging and non-imaging data by combining a CNN with a CatBoost classifier for final disease diagnosis. All models in Ncomms were optimized with AdamW. The CNN for MRI data was trained with a learning rate of 0.001 for 101 epochs. The CatBoost regression models for non-imaging data were trained with a dropout rate of 0.5, a batch size of 32, and a learning rate of 0.001. The fusion CatBoost classifier, integrating imaging and non-imaging inputs, was trained with the same hyperparameters as the non-imaging models (dropout 0.5, batch size 32, learning rate 0.001).

AIdiagnosis proposes a multi-modal ML framework to process a diverse array of clinical textual data as well as multi-modal neuro imaging data to perform disease diagnosis. Model training was performed using a mini-batch strategy with the AdamW optimizer, employing an initial learning rate of 0.001 over 256 total epochs. A cosine annealing learning rate scheduler with warm restarts was used to facilitate convergence, with the first restart occurring at epoch 64 and each subsequent restart period doubled relative to the previous one. The hyperparameters were empirically set as follows: $\varepsilon = 0.25$, $\lambda = 0.005$, and $\beta = 0.0005$.

SMART consists of dual visual–textual branches, including an image encoder for MRI data and a text encoder for clinical records. A gated attention transformer serves as the fusion module, integrating features from both branches for joint representation learning and performing AD diagnosis. The model is trained with the Adam optimizer using a learning rate of 3e-4, batch size of 32, for 300 epochs. The hyperparameters are set as $\alpha = \beta = 1$ and temperature $\tau = 0.05$.

## D.2 MORE UNIMODAL COMPARED BASELINES.

Under the modality-complete setting, we evaluate several strong uni-modal baselines, including Random Forest Rigatti (2017) and LLaMA 2 Touvron et al. (2023) for clinical records, Brain-

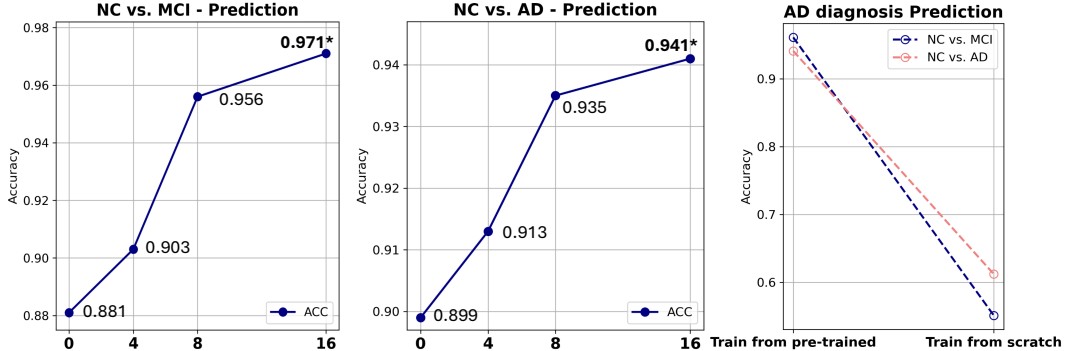

Figure 9: Analysis of how query number in modality-aware Q-former and foundation pre-trained weights affect AD diagnosis performance.

JEPA Dong et al. (2024) and BrainLM Caro et al. (2024) for fMRI, BrainMVP Rui et al. (2025), SAM-Med3D Wang et al. (2023), Swin-UNETR Tang et al. (2022), and M³AE Liu et al. (2023b) for sMRI, as well as NT-Transformer Nguyen et al. (2023), SEI Chen et al. (2022), and DNA-Bert2 Zhou et al. (2024) for genomic data. Table 9 summarizes the performance of uni-modal baselines across three distinct AD-related tasks. Among all modality-specific models, LLaMA achieves the highest overall performance, highlighting the rich and discriminative nature of clinical textual data in Alzheimer's disease analysis. LLaMA, NT, BrainMVP, and BrainJePA consistently outperform other foundation models within their respective modalities, confirming their suitability for integration into our framework.

### D.3 COMPARISON WITH FUSION BASELINES.

Table 10 compares our modality-anchored interaction with three fusion baselines, all built on the same pre-trained foundation models. The baselines include (i) feature concatenation, which directly merges modality features, and (ii) linear or self-attention fusion, which projects modality features into a shared space. Our input-level modality-anchored interaction consistently outperforms these baselines, demonstrating that performing fusion at the input level enables richer cross-modal integration than output-level fusion.

### D.4 EFFECT OF TRAIN FROM PRETRAINED WEIGHTS.

Fig. 9 (rightmost subfigure) compares training from scratch with training from pre-trained weights on two AD diagnosis tasks on ADNI dataset. The results consistently show that initializing from pre-trained weights yields better performance than training from scratch, highlighting the benefit of leveraging prior knowledge encoded in foundation models.

### D.5 VISUALIZATION OF ATTENTION MAPS ON TEXTUAL RECORDS AND sMRI IMAGE

To evaluate interpretability, we compare attention maps produced by our method against those from a late-fusion baseline. Fig 10 (a) presents the attention weight distribution over clinical text descriptions. Results indicate that our model better captures features in longer sequences. For instance, in the sentence 'The APOE status is 0,' a known AD biomarker Knopman et al. (2007), the baseline assigns little attention, whereas our model effectively identifies task-relevant words. Fig 10 (b) compares the sMRI attention maps between our method and the late-fusion baseline. Image patches with the highest attention weights are highlighted in red (late-fusion) and blue (ours). The baseline late fusion method overlooks some critical medical imaging biomarkers, whereas our method more accurately attends to these key subregions.

We further visualize the AD patients' attention maps of sMRI images and clinical records across different anchor modalities and their models in Fig 11. In Stage 2, each anchor modality model takes sMRI or clinical record representations as input. To reveal which regions are most attended,

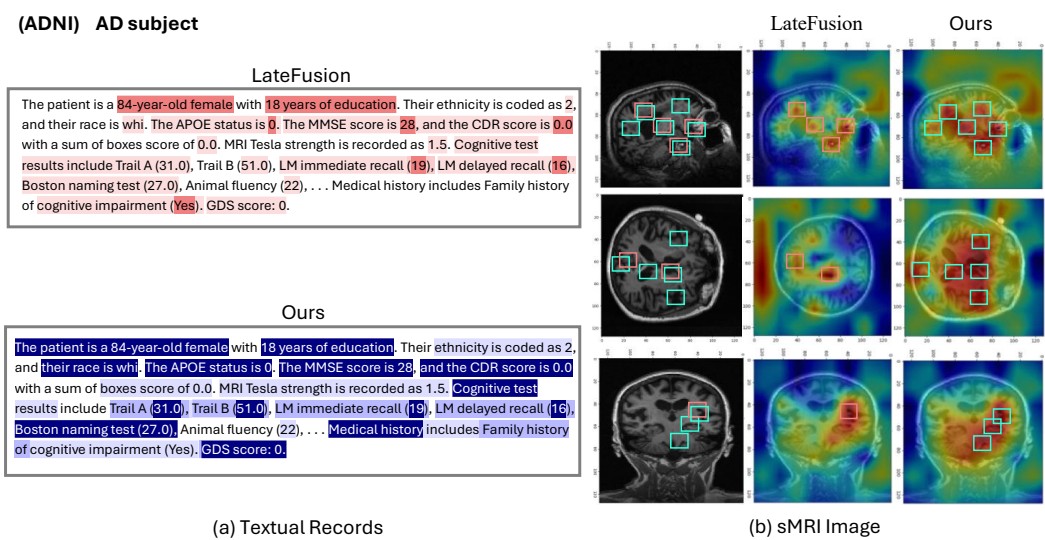

Figure 10: Comparison of attention maps between the late-fusion baseline and our method on clinical textual records and sMRI images of an AD patient from the ADNI dataset. A darker color indicates a higher attention on text. Image patches with the highest attention weights are highlighted in red (late-fusion) and blue (ours).

we compute the attention weights of input data assigned by the model embeddings. As shown in Fig 11, for sMRI data, the fMRI foundation model (BrainJePA) attends to critical medical subregions, while the gene (NT) and clinical text (LLaMA) foundation models cover broader areas. For clinical records, both the gene (NT) and fMRI (BrainJePA) models show broad attention to demographic information, cognitive scores, neuropsychiatric symptoms, medical history, and lifestyle factors, whereas the sMRI foundation model (BrainMVP) focuses mainly on demographic information and neuropsychiatric symptoms. These findings show that our modality-anchored interaction enables each anchor modality to selectively attend to the text and sMRI regions most discriminative for its own semantic representation, and promotes meaningful interactions between modalities, allowing them to complement each other and produce more discriminative multi-modal representations.

### D.6 MULTI-MODAL BIOMARKER ASSOCIATIONS VIA Q-FORMER

We visualize the attention maps of AD patients' sMRI scans across different anchor-modality models trained in Stage 2 (BrainMVP and NT-Transformer), highlighting the brain regions that each model attends to most during prediction, as shown in Fig 12. Specifically, the NT-Transformer trained with genetic features (including APOE 4 status) can highlight hippocampal and medial temporal lobe regions on sMRI, as shown in Fig 12 b. This pattern aligns closely with well-established AD neurobiology: extensive prior studies Li et al. (2016); Bailey et al. (2024) have shown that APOE 4 is strongly associated with hippocampal and parahippocampal atrophy. Therefore, the fact that our gene-anchored model also attends to hippocampal regions on sMRI indicates that the q-former successfully captures meaningful cross-modal biomarker relationships. This provides experimental evidence that our model learns biologically grounded correspondences between different modalities.

### D.7 FURTHER EXPLANATION OF FIGURE 3

In this section, we provide a detailed analysis of Fig. 3, focusing on ablation experiments conducted under the modality-complete setting with various modality combinations. The combinations are grouped into five categories: Neuroimage Unimodal, Non-imaging Unimodal, Clinical Textual Bimodal, Neuroimage Bimodal, and Multi-modal. In the Neuroimage Unimodal group, sMRI and fMRI are evaluated independently to assess their individual predictive capacity. The Non-imaging Unimodal group includes clinical textual records and genomic data, also evaluated separately. For the

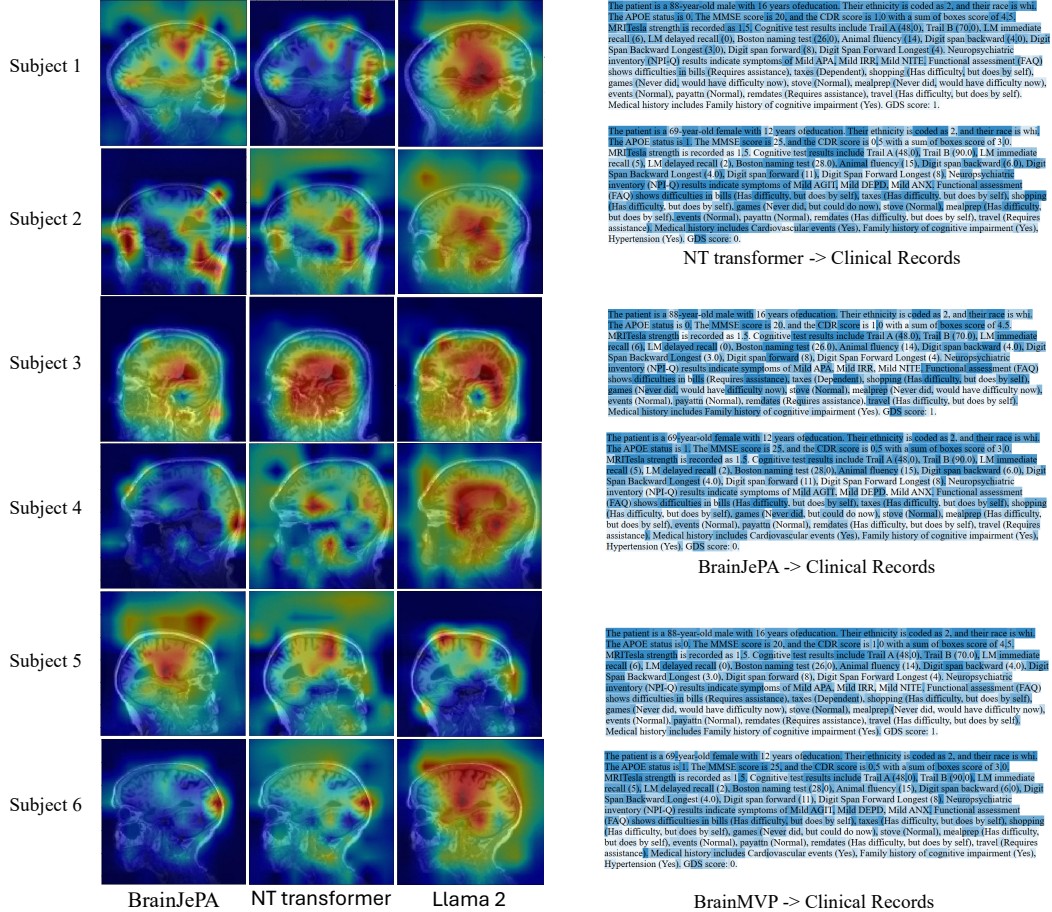

Figure 11: Attention maps of sMRI images and clinical text records across anchor modalities and their models, computed for NC/AD classification on the ADNI dataset.

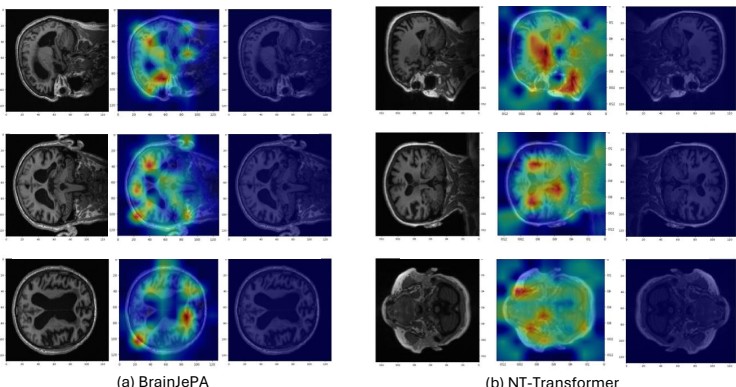

(a) BrainJePA

(b) NT-Transformer

Figure 12: Multi-Modal Biomarker Associations via Q-Former Mechanisms.

Clinical Textual Bimodal group, clinical textual data is paired in turn with each of the other three modalities to assess complementary effects. In the Neuroimage Bimodal group, sMRI and fMRI are each combined with one of the remaining modalities to evaluate how neuroimaging enhances different types of information. Finally, the Multi-modal group incorporates all four modalities to assess the full potential of multi-modal integration. Striped red bars indicate ablation settings where specific

Table 11: Ablation studies on clinical assessment scores (MMSE, MoCA, and CDR) to evaluate their contributions to our framework under the modality-complete setting of the ADNI dataset.

| Methods | NC vs. MCI | | NC vs. AD | | pMCI vs. sMCI | |
|---|---|---|---|---|---|---|
| | ACC | AUC | ACC | AUC | ACC | AUC |
| Random Forest (only scores) | 0.703 ± 0.015 | 0.694 ± 0.012 | 0.738 ± 0.009 | 0.758 ± 0.011 | 0.667 ± 0.023 | 0.665 ± 0.015 |
| Random Forest | 0.709 ± 0.007 | 0.711 ± 0.020 | 0.745 ± 0.007 | 0.768 ± 0.009 | 0.696 ± 0.010 | 0.670 ± 0.012 |
| Ours w/ Scores | **0.871 ± 0.012** | **0.867 ± 0.010** | **0.846 ± 0.015** | **0.854 ± 0.014** | **0.763 ± 0.022** | **0.786 ± 0.027** |
| Ours w/o Scores | 0.850 ± 0.055 | 0.840 ± 0.103 | 0.819 ± 0.105 | 0.822 ± 0.054 | 0.726 ± 0.078 | 0.751 ± 0.027 |

Table 12: Parameter complexity of modality-aware Q-Former with increasing number of modalities.

| Number of Modalities | Modalities | Q-Former Parameters (M) |
|---|---|---|
| 1 | Clinical Records | 9.32 |
| 2 | Clinical Records + Gene | 26.52 |
| 3 | Clinical Records + Gene + sMRI | 45.56 |
| 4 | Clinical Records + Gene + sMRI + fMRI | 63.82 |

modalities are removed from a bi-modal or multi-modal combination to assess its combination contribution. For instance, in the Clinical Textual bimodal group, the red striped bars represent the removal of clinical data. In the Neuroimage bimodal group, the striped bars reflect the exclusion of neuroimaging data. When sMRI and fMRI are combined, the striped bars correspond to removing sMRI to evaluate the standalone impact of fMRI. In the Multi-modal group, the striped bars indicate the exclusion of the G,S and F modalities.

## D.8 SCALABILITY ANALYSIS OF THE MODALITY-AWARE Q-FORMER

As shown in Table 12, compared to the incorporated uni-modal foundation models, which range in size from 500M parameters (NT-Transformer) to 7B parameters (LLaMA), the Q-Former is a relatively lightweight module with only 60M parameters(additional cost compared to late fusion baseline). As such, it does not introduce much computational overhead to the overall framework. Furthermore, the Q-Former's parameter count increases with the number of modalities, as shown in the table below. For instance, it starts with 9.32M parameters for the Clinical records modality alone and increases to 63.82M when all four modalities are included. This demonstrates that the additional computational cost introduced by Q-Former remains manageable as more modalities are integrated.

## D.9 ABLATION STUDIES ON CLINICAL ASSESSMENT SCORES.

We conduct an ablation studies on clinical assessment scores, and results in Table 11 show that clinical assessment scores play an important role in model performance. Our ablation experiments confirm that removing these scores leads to a measurable performance drop. However, the model still achieves strong results even without them, indicating that it effectively leverages broader textual information beyond just the structured assessment scores. Furthermore, our model achieves stronger result andd outperforms the Random Forest baseline significantly, indicating that our method leverages rich textual information beyond clinical scores.

## E THE USE OF LARGE LANGUAGE MODELS (LLMs)

We use LLMs solely for checking grammar and polishing writing.

