# OpenReview forum: "Joint Adaptation of Uni-modal Foundation Models for Multi-modal Alzheimer's Disease Diagnosis"
_ICLR.cc/2026/Conference — ICLR 2026 Poster_

### Official Review · Reviewer_okfp · 2025-10-30

**Soundness:** 2
**Presentation:** 3
**Contribution:** 2
**Rating:** 2
**Confidence:** 4

**Summary:**

This paper proposes a multi-modal framework for AD diagnosis that enables joint interaction among uni-modal foundation models through modality-anchored interaction. The experimental results show the effectiveness of the proposed framework.

**Strengths:**

(1) This paper proposes modality-aware Q-formers that selectively map auxiliary modality features into the anchor model’s feature space, enabling the anchor model to jointly process its own features together with the seamlessly integrated auxiliary features.

 (2) The proposed method is evaluated on AD diagnosis and progression prediction tasks involving the four most common data modalities, and experimental results validate the effectiveness of the proposed framework.

**Weaknesses:**

(1) This paper designates one modality’s foundation model as an anchor and freezes most of its parameters to preserve its feature space, while projecting auxiliary modalities’ features extracted by other foundation models into this space for cross-modal interaction. This technical novelty is very limited.

 (2) There are several existing works about Modality-aware Q-formers (Tong et al., 2024; Zong et al., 2024; Alayrac et al., 2022; Liu et al., 2023a). Thus, it is not clear about the difference between the proposed model with these existing works.

(3) This paper should compare the proposed model with more state-of-the-art AD diagnosis methods.

 (4) The authors should clearly summarize the contributions of this paper.

**Questions:**

(1) There are several existing works about Modality-aware Q-formers (Tong et al., 2024; Zong et al., 2024; Alayrac et al., 2022; Liu et al., 2023a). Thus, it is not clear about the difference between the proposed model with these existing works.

(2) This paper should compare the proposed model with more state-of-the-art AD diagnosis methods.

(3) The authors should clearly summarize the contributions of this paper.

---

> ### Author Response · Authors · 2025-11-21
> **Rebuttal by Authors**
>
> # To Reviewer 4 (okfp)
> We thank you for the suggestions and appreciate comments like “effectiveness”.  We will gladly incorporate all the feedback in the revised version.
>
> ## Technical Novelty of the Framework
> > Q: This paper designates one modality’s foundation model as an anchor and freezes most of its parameters to preserve its feature space, while projecting auxiliary modalities’ features extracted by other foundation models into this space for cross-modal interaction. This technical novelty is very limited.
>
> We respectfully disagree with the reviewer’s assessment regarding the technical novelty. We would like to clarify that our modality-anchored interaction introduces substantial technical novelty beyond existing multimodal fusion approaches, which we summarize from three perspectives:
>
>
> ### Breadth of modality inclusivity
> In this work, we address application scenarios characterized by much **greater modality diversity** and **higher heterogeneity**, which remain largely unexplored in existing research. While powerful uni-modal foundation models exist, prior multimodal approaches (Xue etal.,2024;Chen&Hong,2024;Qiu etal.,2022) typically combine only two or three modalities—often limited to imaging (e.g., MRI) or paired with basic clinical features—using relatively simple fusion strategies.
>
> In contrast, our framework tackles a substantially more challenging setting: integrating four heterogeneous modalities, including high-dimensional time-series signals such as fMRI and sequential data like gene sequence, each paired with its own pretrained foundation model. This scenario reflects real-world clinical complexity, where diverse data sources (imaging, genetics, clinical records, cognitive assessments) must be jointly analyzed for accurate diagnosis and prognosis.
>
> This breadth of modality inclusivity and structured integration has not been modeled in prior multimodal research, marking a significant step toward clinically realistic and generalizable multimodal AI systems.
>
> ### Asymmetric Fusion Framework
> Prior multimodal fusion methods assume that all modalities can be mapped into a shared latent space by concatenating and integrating via attention or linear modules. Even recent attempts such as M4Survive (Lee et al., 2025) mainly adopt symmetric late-fusion schemes, which process modalities independently and combine them only at the decision stage.
>
> However, foundation models for different biomedical modalities are inherently heterogeneous and structurally incompatible, which limits the depth of inter-modal interaction in prior fusion-based approaches.
>
> One key contribution of our modality-anchored interaction is to designate each modality as  anchor modality in turn, explicitly preserving its pretrained feature space from pretrained models while aligning other modalities toward it, rather than collapsing all modalities into a generic shared space. This design preserves the integrity of domain-specific representations and enables deeper cross-modal interaction, a property that existing multimodal fusion methods do not consider.
>
> ### Novel Q-Former: Modality-Aware Integration
> Modality-aware Q-Former is a novel Q-former structure specially designed for our asymetric fusion framework, which is significantly different from conventional Q-former.
>
> Specifically, rather than naively projecting modalities through linear projection layers or simple concatenation, our approach propose a novel modality-aware Q-Former with two complementary query types:
> * **Uni-modal queries** to retain modality-specific information for each auxiliary modality.
> * **Multi‑modal queries** that explicitly model cross‑auxiliary interactions prior to projection into the anchor space.
>
> This dual-query driven, modality-preserving projection enables a structured and task-adaptive integration of heterogeneous auxiliary modalities. This mechanism has not been explored in prior diagnosis fusion studies or general multimodal frameworks.

---

> ### Author Response · Authors · 2025-11-21
> **Rebuttal by Authors**
>
> ## Distinguishing Our Modality-Aware Q-Former from Prior Works
> > Q: There are several existing works about Modality-aware Q-formers (Tong et al., 2024; Zong et al., 2024; Alayrac et al., 2022; Liu et al., 2023a). Thus, it is not clear about the difference between the proposed model with these existing works.
>
> ### Listed prior works do not use Modality-aware Q-formers
>
> We would like to clarify a misunderstanding regarding Modality-aware Q-formers. The concept of a Modality-aware Q-former is first introduced in our work and is not employed in any of the cited studies.
>
> The works listed by the reviewer (Tong et al., 2024; Zong et al., 2024; Alayrac et al., 2022; Liu et al., 2023a) propose various multimodal alignment strategies, but none apply Q-formers for modality-aware integration. Specifically, they propose various alignment strategies that map non-text modality into the text embedding space of an LLM, which have been highly inspiring to our work, but they do not use learnable queries or query transformers to extract knowledge.
>
> For example, Tong et al., 2024 proposed SVA, Zong et al., 2024 proposed MoV-Adapter, and Liu et al., 2023 proposed Projection, which are connector-based designs that directly convert visual embeddings into LLM token space. Alayrac et al., 2022 proposed GATED XATTN-DENSE layers, which are inserted between frozen LLM layers to condition the LLM on visual inputs,  rather than using a Q-former mechanism.
>
> ### Novelty of Modality-aware Q-former
> Our Modality-aware Q-former introduces architectural and functional innovations that go beyond prior Q-former designs. Specifically, its novelty lies in three key aspects:
>
> * **Single-target projection vs. general projection**.  Prior works employ Q‑formers[1,2] primarily as *single‑target projection* modules that map each non‑text modality into the text embedding space of an LLM. By contrast, our modality‑anchored design sequentially treats each modality as the anchor and projects the remaining modalities into that anchor’s representation space. This yields a general projection mechanism that is not restricted to text and supports direct alignment among any pair of modalities (e.g., fMRI↔clinical, sMRI↔genetics), thereby enabling richer supervision signals and task‑aligned representational geometry.
>
> * **Auxiliary‑modality interactions before projection.** Prior Q‑formers typically compress each modality independently and only “meet” at the LLM or a late‑fusion head[3,4], which limits cross‑modal reasoning. Our modality‑aware Q‑former introduces two complementary query types:
>   * **Uni‑modal queries** that preserve modality‑specific information for each auxiliary modality, and
>   * **Multi‑modal queries** that explicitly model cross‑auxiliary interactions prior to projection into the anchor space.This design lets auxiliary modalities mutually inform one another and produces fused features that are more discriminative and clinically coherent.
>
> * **Head-to-head performance comparison.** In head‑to‑head comparisons against a BLIP‑style Q‑former[1] under the same datasets, backbones, and training settings, our architecture consistently outperforms—indicating that the gains stem from the proposed mechanisms rather than domain idiosyncrasies.
>
> | **Methods**                  | **NC vs. MCI (ACC ↑)** | **NC vs. AD (ACC ↑)** | **sMCI vs. pMCI (ACC ↑)** | **Age (MSE ↓)** | **MMSE (MSE ↓)** |
> |------------------------------|-------------------------|------------------------|----------------------------|------------------|-------------------|
> | **Modality-aware Q-former**      | **0.979 ± 0.004**          | **0.945 ± 0.011**         | **0.846 ± 0.005**             | **0.241 (.017)**    | **0.397 (.015)**     |
> | BLIP2 Query Transformer      | 0.960 ± 0.003          | 0.921 ± 0.003         | 0.831 ± 0.009             | 0.311 (.015)    | 0.402 (.018)     |
>
>
> The results show that our design consistently outperforms Query transformer from BLIP2 on all evaluated tasks, confirming the observed gains from our proposed modality-aware Q-Former.

---

> ### Author Response · Authors · 2025-11-21
> **Rebuttal by Authors**
>
> ## Comparison with Additional Baselines
> > Q: This paper should compare the proposed model with more state-of-the-art AD diagnosis methods.
>
> Our original experiments in Tables 1 and 2 already include a wide range of recently published and representative AD diagnosis methods, spanning uni-modal, bi-modal, and multi-modal approaches. These methods cover strong-performing uni-modal models for our four modalities (clinical data, sMRI, fMRI, and genetic data), as well as methods combining sMRI and clinical data (e.g., NComms, AI-Diagnosis, and SMART) and recent fusion-based strategies such as M4Survive. To further address the reviewer’s concern, we additionally compared our framework with more multimodal diagnosis models that integrate sMRI and clinical data, including Irene[5], VAP-Former[6], and Alifuse[7]. The results are summarized below:
>
> | **Methods**    | **NC vs. MCI (ACC / SPE / SEN)** | **NC vs. AD (ACC / SPE / SEN)** | **pMCI vs. sMCI (ACC / SPE / SEN)** |
> |---------------|-----------------------------------|----------------------------------|--------------------------------------|
> | Irene         | 0.892 / 0.905 / 0.855            | 0.903 / 0.942 / 0.867           | 0.759 / 0.732 / 0.746               |
> | VAP-Former    | 0.910 / 0.923 / 0.868            | 0.905 / 0.932 / 0.875           | 0.775 / 0.787 / 0.729               |
> | Alifuse       | 0.913 / 0.929 / 0.866            | 0.899 / 0.937 / 0.869           | 0.783 / 0.792 / 0.740               |
> | **Ours**      | **0.979 / 0.957 / 0.963**        | **0.945 / 0.960 / 0.931**       | **0.846 / 0.901 / 0.711**           |
>
> These results show that our framework consistently outperforms those multimodal baselines across all three classification tasks, further confirms the superiority and robustness of our approach in multimodal AD diagnosis scenarios.
>
> ## Summary of Contributions
> > Q: The authors should clearly summarize the contributions of this paper.
>
> We have provide a detailed analysis on the contributions and novelty of this paper in the previous responses. We appreciate the reviewer’s request for a clear summary of our contributions. Here we summerize these analysis into three key advances.
>
> * **Leveraging foundation models for AD diagnosis:** We are among the first to integrate multiple uni-modal foundation models (for sMRI, fMRI, clinical records, and genetic data) into a unified framework, enabling robust transfer from large-scale pretraining and addressing the scarcity of labeled medical data. This design achieves state-of-the-art performance and strong generalization across external datasets and even other neurodegenerative diseases.
>
> * **Modality-anchored interaction:** Unlike conventional fusion strategies that jointly train or naively align heterogeneous representations, we propose a modality-anchored mechanism that iteratively designates each modality’s as anchor one in turn, and projects auxiliary modalities into anchor feature space for interaction. By applying this process to all modalities, our framework achieves deep cross-modal integration, preserves modality-specific strengths, and scales effectively to four foundation models across sMRI, fMRI, clinical records, and genetic data.
>
> * **Modality-aware Q-formers** that use learned queries to extract relevant auxiliary information and project it into the anchor space, allowing fine-grained alignment and joint processing without compromising the strengths of each foundation model. Together, these contributions provide a principled solution to the core challenge of integrating heterogeneous foundation models for multimodal AD diagnosis, which prior works have not addressed.
>
> ## Reference
> [1]Li, Junnan, et al. "Blip: Bootstrapping language-image pre-training for unified vision-language understanding and generation." ICML 2022.
>
> [2]Li, Junnan, et al. "Blip-2: Bootstrapping language-image pre-training with frozen image encoders and large language models." International conference on machine learning. PMLR, 2023.
>
> [3]Dai, Wenliang, et al. "Instructblip: Towards general-purpose vision-language models with instruction tuning." Advances in neural information processing systems 36 (2023): 49250-49267.
>
> [4]Yeo, Jeong Hun, et al. "MMS-LLaMA: Efficient LLM-based Audio-Visual Speech Recognition with Minimal Multimodal Speech Tokens." arXiv preprint arXiv:2503.11315 (2025).
>
> [5]Zhou, Hong-Yu, et al. "A transformer-based representation-learning model with unified processing of multimodal input for clinical diagnostics." Nature biomedical engineering 7.6 (2023): 743-755.
>
> [6]Kang, Luoyao, et al. "Visual-attribute prompt learning for progressive mild cognitive impairment prediction." International Conference on Medical Image Computing and Computer-Assisted Intervention. Cham: Springer Nature Switzerland, 2023.
>
> [7]Chen, Qiuhui, and Yi Hong. "Alifuse: Aligning and Fusing Multimodal Medical Data for Computer-Aided Diagnosis." BIBM. 2024.

---

> > ### Author Response · Authors · 2025-11-28
> >
> > Based on your review, we have added several new experiments and clarifications:
> >
> > * Ablation study comparing our modality-aware Q-former with the BLIP Q-former under the same setting.
> > * Additional baseline comparisons, including Irene, VAP-Former, and Alifuse.
> > * Further clarification on our modality-anchored interaction mechanism and modality-aware Q-former design.
> > * A concise summary of our contributions.
> >
> > If you have a moment to review our updates, we would be grateful for any follow-up questions before the deadline.

---

### Official Review · Reviewer_gHU6 · 2025-10-31

**Soundness:** 4
**Presentation:** 4
**Contribution:** 4
**Rating:** 8
**Confidence:** 5

**Summary:**

This paper presents a novel multi-modal framework for diagnosing Alzheimer's Disease (AD) through the joint adaptation of uni-modal foundation models. The central concept is the utilization of a "modality-anchored interaction" strategy. In this strategy, the foundation model of one modality serves as an anchor, and features from other auxiliary modalities are projected into its feature space via a proposed "Modality-aware Q-former". This enables interaction among modalities while maintaining the pre-trained representations of the anchor model. The framework is evaluated in the context of AD diagnosis and progression prediction, leveraging four modalities: sMRI, fMRI, clinical records, and genetic data, and it showcases superior performance compared to previous approaches.

**Strengths:**

- The proposed modality-anchored interaction framework for integrating uni-modal foundation models is a novel and interesting approach.
- The paper addresses the important problem of multi-modal AD diagnosis, which has significant clinical relevance. The use of foundation models is a promising direction.
- The method shows strong performance on AD diagnosis and progression prediction, outperforming several baselines. The generalization experiments on external datasets are a key strength.

**Weaknesses:**

- The proposed framework is quite complex, involving multiple foundation models, Q-formers, and a two-stage training process. This might make it difficult to reproduce and apply in practice.
- While the paper provides some analysis, more in-depth ablation studies could further clarify the contribution of each component (e.g., the cross-modal Q-former, the LoRA fine-tuning).
- The paper states that each modality is used as an anchor in turn, and the final prediction is an aggregation. It would be interesting to see an analysis of how the choice of anchor modality affects performance. Is there an optimal anchor?

**Questions:**

1.  Could you provide more details on the computational cost of the proposed framework, both during training and inference?
2.  Have you experimented with other methods for aggregating the predictions from the different anchor models? For example, a weighted average based on the confidence of each model.
3.  How sensitive is the model to the number of learnable queries in the Q-formers?

---

> ### Author Response · Authors · 2025-11-21
> **Rebuttal by Authors**
>
> # To Reviewer 3 (gHU6)
> We thank you for the suggestions and appreciate comments like “novel and interesting approach”, “addresses the important problem”, “promising direction”, “strong performance”, and “a key strength”.  We will gladly incorporate all the feedback in the revised version.
>
> ## Complexity and Practical Applicability
> > Q: The proposed framework is quite complex, involving multiple foundation models, Q-formers, and a two-stage training process. This might make it difficult to reproduce and apply in practice.
>
> We appreciate the reviewer’s comments regarding complexity and practical applicability. To promote transparency and reproducibility, we provide code in the Supplementary Material, and a full open-access version will be released upon publication. All details of the training setup, data splits, hyperparameters, and the two-stage training process are documented in Appendix B.1 (Settings of Experiments) and B.2 (Settings and Hyperparameters) in the original submission.
> * **Computational cost during training and inference.**  Inference with the final model is lightweight: it requires only 381 ms per subject on a single A100 GPU, with 23.4 GB (23,416 MB) of GPU memory consumption. Training of our two-stage pipeline requires 30 A100-GPU hours, with a training throughput of 6.85 samples/s.
>
> Regarding practical feasibility, while our experiments were conducted on A100 GPUs for efficiency, we have verified that the entire training pipeline is feasible on A6000 GPU, requiring 70 A6000-GPU hours. This shows that our method is implementable without requiring extreme computational resources. We hope these measures can ensure that our framework is both reproducible and applicable in practice.
>
> ## In-depth Ablation Studies
> > Q: While the paper provides some analysis, more in-depth ablation studies could further clarify the contribution of each component (e.g., the cross-modal Q-former, the LoRA fine-tuning).
>
> We appreciate the reviewer’s suggestion for more in-depth ablation analysis. To address this, we conducted additional ablation studies on the components of our modality-aware Q-Former, which consists of Uni-modal Q-formers and Cross-modal Q-formers. The accuracy results are provided in the table below.
>
> | **Components**               | **NC vs. MCI** | **NC vs. AD** | **sMCI vs. pMCI** |
> |------------------------------|---------------|--------------|-------------------|
> | w/ Uni-modal Q-formers       | 0.956         | 0.911        | 0.801             |
> | w/ Cross-modal Q-formers     | 0.947         | 0.919        | 0.827             |
> | **Modality-aware Q-former**      | **0.979**     | **0.945**    | **0.846**         |
>
> Results indicate that Cross-modal Q-formers enhance interaction compared to Uni-modal alone, while combining both components achieves the best performance across all tasks. This confirms their complementary roles: Uni-modal Q-formers reinforce modality-specific representations, whereas Cross-modal Q-formers enable effective integration of auxiliary modalities and produce fused features that are more discriminative and clinically coherent.
>
>
> In addition to analyzing Q-former components, we conducted an ablation study on LoRA fine-tuning ranks to evaluate its impact on performance. The accuracy results are shown below:
>
> | **Rank** | **NC vs. MCI** | **NC vs. AD** | **sMCI vs. pMCI** |
> |----------|---------------|--------------|-------------------|
> | R = 1    | 0.793         | 0.779        | 0.688             |
> | R = 2    | 0.861         | 0.859        | 0.764             |
> | R = 4    | 0.935         | 0.911        | 0.816             |
> | R = 16   | 0.962         | 0.928        | 0.833             |
> | **R = 32**   | **0.979**     | **0.945**    | **0.846**         |
>
> These results show that increasing LoRA rank improves performance, with R = 32 achieving the best results across all tasks. Therefore, sufficient adaptation capacity is critical for leveraging foundation models effectively.

---

> > ### Author Response · Authors · 2025-11-21
> > **Rebuttal by Authors**
> >
> > ## Choice of Anchor Modality during Aggregation
> > > Q: The paper states that each modality is used as an anchor in turn, and the final prediction is an aggregation. It would be interesting to see an analysis of how the choice of anchor modality affects performance. Is there an optimal anchor?
> >
> > We thank the reviewer for the insightful suggestion. To directly examine whether the choice of anchor modality affects performance and whether an optimal anchor exists, we conducted an additional experiment following the reviewer’s suggestion. Specifically, we use the trained framework to obtain the four anchor models and evaluated each of them. Each anchor modality model (sMRI (S), fMRI (F), gene (G) and clinical text (C)) generates its own prediction through the pipeline. This allows us to isolate the diagnostic capability uniquely contributed by each anchor modality.  The accuracy results are in Table below.
> >
> > | **Anchor(s)**       | **MCI vs. NC** | **AD vs. NC** | **pMCI vs. sMCI** |
> > |----------------------|---------------|--------------|-------------------|
> > | S                   | 0.905         | 0.877        | 0.813             |
> > | F                   | 0.936         | 0.913        | 0.806             |
> > | G                   | 0.887         | 0.897        | 0.799             |
> > | C                   | 0.944         | 0.929        | 0.811             |
> > | C+S                 | 0.948         | 0.931        | 0.823             |
> > | C+G                 | 0.941         | 0.922        | 0.826             |
> > | C+F                 | 0.963         | 0.931        | 0.825             |
> > | S+G                 | 0.913         | 0.898        | 0.804             |
> > | F+G                 | 0.929         | 0.921        | 0.830             |
> > | S+F                 | 0.932         | 0.931        | 0.819             |
> > | **C+G+S+F**         | **0.979**     | **0.945**    | **0.846**         |
> >
> > Our experiment results show that clinical text ( C ) and fMRI (F) stand out as strong anchors, delivering higher accuracy than sMRI or genetic data, whether evaluated individually or in combination. However, neither modality consistently dominates across all tasks. Instead, combining anchors improves performance significantly, as seen in pairs like C+F, which outperform any single anchor. This trend culminates in full multi-modal integration, where aggregating predictions from all four anchors achieves the best results across all metrics. These findings validate our design choice: rather than relying on one anchor, our framework leverages complementary strengths from all modalities.
> >
> > ## Aggregation Strategies
> > > Q: Have you experimented with other methods for aggregating the predictions from the different anchor models? For example, a weighted average based on the confidence of each model.
> >
> > We thank the reviewer for the insightful suggestion regarding exploring other aggregation strategies beyond simple averaging. Following the reviewer’s recommendation, we conducted an additional experiment using a confidence weighted averaging strategy across the four anchor models. The weighting coefficient for each anchor was computed from its confidence. The results are shown below:
> >
> > | **Method**                                | **NC vs. MCI** | **NC vs. AD** | **pMCI vs. sMCI** |
> > |-------------------------------------------|---------------|--------------|-------------------|
> > | C+G+S+F (ours)       | 0.979         | **0.945**       | 0.846             |
> > | C+G+S+F (weighted aggregation)            | **0.981**     | 0.944        | **0.847**         |
> >
> > While weighted aggregation is feasible, it yields a slight improvement, showing that our framework is robust to the choice of aggregation strategy.
> >
> > ## Sensitivity to Number of Queries
> > > Q: How sensitive is the model to the number of learnable queries in the Q-formers?
> >
> > We thank the reviewer for raising this important question about the sensitivity to the number of queries. In the original submission (Figure 5), we reported an analysis showing that 16 queries achieved the best accuracy. To further address the reviewer’s concern, we conducted an extended sweep over a wider range of query counts {0, 4, 8, 16, 32}.
> >
> > | **Query Number**       | **0**   | **4**   | **8**   | **16**  | **32**  |
> > |------------------|--------|--------|--------|--------|--------|
> > | **NC vs. MCI**   | 0.881  | 0.903  | 0.956  | **0.971**  | 0.969  |
> > | **NC vs. AD**    | 0.899  | 0.913  | 0.935  | **0.941**  | 0.940  |
> >
> > Results above shows our framework is not highly sensitive to the number of learnable queries. A lightweight configuration with only 8 queries already provides strong performance, whereas 16 queries offer the best trade-off between accuracy and computational efficiency.

---

### Official Review · Reviewer_x1Jq · 2025-10-31

**Soundness:** 3
**Presentation:** 2
**Contribution:** 2
**Rating:** 4
**Confidence:** 4

**Summary:**

This paper proposes to use multiple Q-formers to align various medical modalities for Alzheimer's disease prediction. The authors specifically use train these Q-formers by taking one modality as an anchor, and aligning the other modalities to the latent space of the anchored modality.

**Strengths:**

The quantitative results are quite convincing, the authors compare the method across three different datasets, and they perform various ablation studies.

**Weaknesses:**

1) The authors focus a lot on classification performance, but there are many other tasks that are possible within these datasets. Specifically, to further strengthen their results the authors can show that their model also performs better when predicting biomarkers, age, etc. The classification tasks, although important, are not the only way to evaluate these models.
2) The approach is not that technically novel. There are many papers that align modalities using Q-formers, and it is therefore unclear to me what the technical novelty is in this paper beyond a new application area. A paper that mostly focuses on a new application area is fine, but the application area is quite limited (the authors restrict themselves to Alzheimer's disease instead of broadly neurological disorders), and although results are clearly better in the tables, it is unclear how significant these results are.
3) This brings me to the third point, which is that the authors do not seem to use cross-validation when evaluating their model(s), and do not compute confidence intervals or standard deviations for their results. Especially in neuroimaging, where the exact training and test subsets can have a large impact on the results, it is important to perform experiments over multiple training and test sets, and even initialization seeds, to ensure results are repeatable.
4) The authors mention modality-aware Q-formers in the introduction, but do not discuss the referenced papers in depth in the related work section. In general, I think the authors can do a much better job at placing their work into the context of current work, especially given how large the field of multimodal foundation models is. Make sure to highlight exactly how your model and use of Q-formers is different, and why this is leading to performance improvements.

**Questions:**

1) Figure 2: What is the difference between the authors' method and a Q-former?
2) The unimodal performance for the M4Survive and LateFusion models is sometimes worse than the unimodal performance, which begs the question: How fair are the comparisons with the baselines? Do the authors use the same unimodal foundation models and fine-tuning approaches? For example, in the authors' model they use fine-tuned unimodal foundation models, is this also the case for the multimodal baselines?

---

> ### Author Response · Authors · 2025-11-21
> **Rebuttal by Authors**
>
> # To Reviewer 2 (x1Jq)
> We thank you for the constructive suggestions and appreciate comments like “convincing quantitative results”. We will gladly incorporate all the feedback in the revised version.
>
> ## Additional tasks beyond classification
> > Q: The authors focus a lot on classification performance, but there are many other tasks that are possible within these datasets. Specifically, to further strengthen their results the authors can show that their model also performs better when predicting biomarkers, age, etc. The classification tasks, although important, are not the only way to evaluate these models.
>
> We thank you for noting that classification tasks are not the only way to evaluate our models. Following your suggestion, we additionally conducted experiments on two regression tasks predicting clinically relevant continuous variables: age and MMSE scores, using both MSE and Pearson correlation (ρ) as evaluation metrics.
>
>
> | **Methods**     | **Age ( MSE ↓)** | **Age ( ρ ↑)** | **MMSE ( MSE ↓)** | **MMSE ( ρ ↑)** |
> |------------------|-----------------|---------------|-------------------|-----------------|
> | Ncomms           | 0.414 (.017)   | 0.731 (.023)  | 0.493 (.026)      | 0.699 (.021)    |
> | AI-diagnosis     | 0.396 (.018)   | 0.775 (.027)  | 0.412 (.011)      | 0.732 (.030)    |
> | SMART            | 0.406 (.015)   | 0.743 (.034)  | 0.612 (.017)      | 0.497 (.023)    |
> | M4Survive        | 0.582 (.021)   | 0.613 (.024)  | 0.542 (.020)      | 0.587 (.012)    |
> | **Ours**         | **0.241 (.017)** | **0.874 (.023)** | **0.397 (.015)** | **0.791 (.027)** |
>
> As shown above, our model achieves the lowest MSE and the highest correlation across both regression tasks, outperforming all baselines by a clear margin. These additional results show that our framework can perform well in both classification and regression tasks, and is capable of supporting broader clinical applications beyond disease diagnosis.

---

> > ### Author Response · Authors · 2025-11-21
> > **Rebuttal by Authors**
> >
> > ## Technical Novelty and Application Scope
> > > Q: The approach is not that technically novel. There are many papers that align modalities using Q-formers, and it is therefore unclear to me what the technical novelty is in this paper beyond a new application area. A paper that mostly focuses on a new application area is fine, but the application area is quite limited (the authors restrict themselves to Alzheimer's disease instead of broadly neurological disorders), and although results are clearly better in the tables, it is unclear how significant these results are.
> >
> > ### On technical novelty of modality-aware Q-former
> >
> > We respectfully disagree with the reviewer’s assessment regarding the technical novelty of our proposed modality-aware Q-former. Our method introduces architectural and functional innovations that go beyond prior Q-former designs. Specifically, its novelty lies in four key aspects:
> >
> > * **Single-target projection vs. general projection**.  Prior works employ Q‑formers[2,3] primarily as *single‑target projection* modules that map each non‑text modality into the text embedding space of an LLM. By contrast, our modality‑anchored design sequentially treats each modality as the anchor and projects the remaining modalities into that anchor’s representation space. This yields a general projection mechanism that is not restricted to text and supports direct alignment among any pair of modalities (e.g., fMRI↔clinical, sMRI↔genetics), thereby enabling richer supervision signals and task‑aligned representational geometry.
> > * **Auxiliary‑modality interactions before projection.** Prior Q‑formers typically compress each modality independently and only “meet” at the LLM or a late‑fusion head[5,8], which limits cross‑modal reasoning. Our modality‑aware Q‑former introduces two complementary query types:
> >   * **Uni‑modal queries** that preserve modality‑specific information for each auxiliary modality, and
> >   * **Multi‑modal queries** that explicitly model cross‑auxiliary interactions prior to projection into the anchor space.This design lets auxiliary modalities mutually inform one another and produces fused features that are more discriminative and clinically coherent.
> >
> > * **Building block of the foundamentally new asymmetric fusion paradigm.** Modality-aware Q-former serves as a building block for the proposed **asymmetric**, **modality-anchored** fusion paradigm, which introduces a fundamentally new approach to multimodal integration rather than a  architectural improvement. Significantly different from the prior works that map all modalities into a single latent space, our paradigm  designates each modality as an anchor in turn, explicitly preserving its pretrained feature space while aligning other modalities toward it. This approach maintains domain-specific integrity and enables deeper cross-modal interaction—an aspect overlooked by existing multimodal fusion methods.
> >
> > * **Head-to-head performance comparison.** In head‑to‑head comparisons against a BLIP‑style Q‑former[2] under the same datasets, backbones, and training settings, our architecture consistently outperforms—indicating that the gains stem from the proposed mechanisms rather than domain idiosyncrasies.
> >
> > | **Methods**                  | **NC vs. MCI (ACC ↑)** | **NC vs. AD (ACC ↑)** | **sMCI vs. pMCI (ACC ↑)** | **Age (MSE ↓)** | **MMSE (MSE ↓)** |
> > |------------------------------|-------------------------|------------------------|----------------------------|------------------|-------------------|
> > | **Modality-aware Q-former**      | **0.979 ± 0.004**          | **0.945 ± 0.011**         | **0.846 ± 0.005**             | **0.241 (.017)**    | **0.397 (.015)**     |
> > | BLIP2 Query Transformer      | 0.960 ± 0.003          | 0.921 ± 0.003         | 0.831 ± 0.009             | 0.311 (.015)    | 0.402 (.018)     |
> >
> > The results show that our design consistently outperforms Query transformer from BLIP2 on all evaluated tasks, confirming the observed gains from our proposed modality-aware Q-Former.

---

> ### Author Response · Authors · 2025-11-21
> **Rebuttal by Authors**
>
> ### On application scope
>
> We appreciate the reviewer’s concern regarding the application scope. We would like to clarify that our paper have demonstrated that the proposed method is not limited to Alzheimer’s Disease but is broadly applicable to neurological disorders and even multi-modal disease modeling in general.
>
> As evidence, Table 3 in the original manuscript already  demonstrates that, in addition to evaluations on AD using ADNI and OASIS datasets, our framework has been successfully applied to Parkinson’s disease diagnosis and progression prediction using PPMI dataset. These results confirm that the proposed approach generalizes beyond a single condition and is adaptable to diverse clinical domains.
>
> | **Modality**        | **Methods**     | **C-Index**         |
> |-----------------|-------------|-----------------|
> | CT              | CT-FM       | 0.723 ± 0.013   |
> | Textual         | LLaMA       | 0.697 ± 0.027   |
> | CT + Textual    | lateFusion  | 0.730 ± 0.011   |
> | **CT + Textual**    | **Ours**        | **0.776 ± 0.020**   |
>
> To further demonstrate generalizability beyond neuro-related tasks, we conducted an additional experiment on a non-neuro domain, the TCGA Kidney survival prediction task, using CT imaging (CT-FM[1]) and clinical text data encoded by LLaMA as inputs. Following the prior survival prediction setup [9], we evaluated methods using the C-index measure. The results show that our framework continues to perform strongly, indicating that it can be readily extended to broader disease-modeling task.
>
>
>
>
> ### On significant of the results
> The significance of the performance improvement our method can be quantified in two aspects. Firstly, compared to the strongest baselines (AI-diagnosis), our method consistently improves performance by significant margin across most metrics. Specifically, for NC vs. MCI, ACC increases by 2.9%, SPE by 2.0%, and SEN by 0.8%. For NC vs. AD, ACC and SPE improves by 2.1%, and SEN by 3.6%. For sMCI vs. pMCI, ACC rises by 2.1% and SPE by 5.2%. To ensure these improvements are statistically significant and robust, we employ 5-fold cross-validation, repeated runs, and paired statistical tests (p = 0.005). We refer the reviewer to our subsequent response for full details, which further demonstrate the significance of these gains.
>
>
> ## Evaluation Significance and Robustness
> > Q: ... the authors do not seem to use cross-validation when evaluating their model(s), and do not compute confidence intervals or standard deviations for their results. ... It is important to perform experiments over multiple training and test sets, and even initialization seeds, to ensure results are repeatable.
>
> We thank the reviewer for raising this important point. We would like to clarify that we do use cross-validation, repeated runs, and compute statistical significance (p-values) to ensure the robustness of our comparisons. These details are already documented in Appendix B.1 (Settings of Experiments) in the original submission.
>
> Specifically, the dataset is partitioned into 60% for training, 20% for validation, and 20% for testing. We conducted test experiments using 5-fold cross-validation. For each fold, we record the performance on the test set. This resulted in five paired performance values, one for each fold. We then performed paired t-test on these fold-wise results to assess the statistical significance of the performance difference. The resulting p-value below 0.01 (p = 0.005) confirms that the performance gains achieved by our method are statistically significant. Our main tables report average performance across cross-validation folds, and we additionally provide the standard deviations for our results in the table below.
>
> | Method       | NC vs. MCI         | NC vs. AD          | sMCI vs. pMCI       |
> |--------------|---------------------|----------------------|----------------------|
> | Ncomms       | 0.945 ± 0.010       | 0.928 ± 0.012        | 0.773 ± 0.013        |
> | AI-diagnosis | 0.950 ± 0.009       | 0.924 ± 0.004        | 0.825 ± 0.008        |
> | SMART        | 0.932 ± 0.013       | 0.917 ± 0.010        | 0.810 ± 0.012        |
> | M4Survive    | 0.926 ± 0.005       | 0.911 ± 0.008        | 0.812 ± 0.007        |
> | LateFusion   | 0.881 ± 0.008       | 0.899 ± 0.010        | 0.801 ± 0.009        |
> | **Ours**     | **0.979 ± 0.004**   | **0.945 ± 0.011**    | **0.846 ± 0.005**    |

---

> ### Author Response · Authors · 2025-11-21
> **Rebuttal by Authors**
>
> ## Discussing Q-formers in related work section
> > Q: ... but author do not discuss the referenced papers in depth in the related work section. In general, I think the authors can do a much better job at placing their work into the context of current work ... Make sure to highlight exactly how your model and use of Q-formers is different, and why this is leading to performance improvements.
>
> We appreciate the reviewer’s suggestion to better contextualize our work within the field of multimodal foundation models. In our response under "Technical Novelty and Application Scope", we have clarified how our model and use of Q-formers differ from prior works, and why this leads to improved performance.
>
> Additionally, we have added a dedicated subsection in Related Work (Section 2: Q-formers in Multi-modal Pretrained Models) to explicitly discuss the differences between our approach and prior Q-former based methods. Details are provided below:
>
> > Prior studies primarily used query transformers (q-formers) [2] or connectors(Liu etal.,2023a) to project non-text modalities, such as images, video, or audio, into the text embedding space of large language models (LLMs) for multimodal alignment. For example, BLIP-2 [3] and MiniGPT-4 [4] use query transformer to extract features from image patches and output query embeddings that the LLM consumes. InstructBLIP [5] extend this approach to fuse images, video, and audio, with separate query transformers for each modality projecting their features into the LLM text space. Similarly, speech-, video-, and audio-visual models like EmoQ [6], Video-LLaMA[7], and MMS-LLaMA[8] employ query transformer to compress their own modality embeddings into textual representations for LLM processing.  Differing from those prior works, where query transformers project each modality exclusively into the text embedding space of LLMs, our modality-anchored interaction sequentially treats each modality as the anchor, with the remaining three modalities serving as auxiliary modalities. Consequently, our Q-former is designed to be more general, capable of projecting into any of the four modality spaces when designated as the anchor, rather than being restricted to text.
>
> ## Fairness of Baseline Comparisons
> > Q: The performance for the M4Survive and LateFusion models is sometimes worse than the unimodal performance ... How fair are the comparisons with the baselines? Do the authors use the same models and fine-tuning approaches? ...
>
> We appreciate the reviewer’s concern regarding the fairness of unimodal vs. multimodal comparisons in Table 1. We would like to clarify that all models, our and all baselines, use exactly the same finetuned unimodal foundation models and the same fine-tuning protocol, as detailed in Appendix B (ours), Appendix D.1 (all compared baselines).
>
> The observed performance drop for M4Survive and LateFusion does not stem from unequal model capacity or training settings. Instead, this outcome is driven by inherent characteristics of the Alzheimer’s disease domain. fMRI and clinical data are highly predictive of AD diagnosis, since clinical data captures cognitive and behavioral deficits, and fMRI reflects functional alterations. So their uni-modal models naturally achieve strong performance. Moreover, the samples in modality-complete setting is relatively small, and fusing high-dimensional multi-modal features easily leads to overfitting, which explains why M4Survive and Late Fusion sometimes drop below unimodal performance. These observations further highlight the necessity of a stronger interaction mechanism when training data is limited, as fusion models struggle to provide gains in this setting.
>
> ## Reference
> [1]Pai, Suraj, et al. "Vision foundation models for computed tomography." arXiv (2025).
>
> [2]Li, Junnan, et al. "Blip: Bootstrapping language-image pre-training for unified vision-language understanding and generation." ICML 2022.
>
> [3]Li, Junnan, et al. "Blip-2: Bootstrapping language-image pre-training with frozen image encoders and large language models." International conference on machine learning. PMLR, 2023.
>
> [4]Zhu, Deyao, et al. "Minigpt-4: Enhancing vision-language understanding with advanced large language models." arXiv preprint arXiv:2304.10592 (2023).
>
> [5]Dai, et al. "Instructblip: Towards general-purpose vision-language models with instruction tuning." Advances in neural information processing systems 36 (2023): 49250-49267.
>
> [6]Yang, et al. "EmoQ: Speech Emotion Recognition via Speech-Aware Q-Former and Large Language Model." arXiv  (2025).
>
> [7]Zhang, et al. "Video-llama: An instruction-tuned audio-visual language model for video understanding." arXiv (2023).
>
> [8]Yeo, et al. "MMS-LLaMA: Efficient LLM-based Audio-Visual Speech Recognition with Minimal Multimodal Speech Tokens." arXiv (2025).
>
> [9]Jaume, et al. "Modeling dense multimodal interactions between biological pathways and histology for survival prediction." CVPR 2024.

---

> > ### Author Response · Authors · 2025-11-28
> >
> > Thank you again for your constructive feedback on our submission. We're writing to follow up with the rebuttal, in which we aimed to address your points directly. The key updates include:
> >
> > * Additional experiments on predicting biomarkers.
> > * Clarifications on our modality-aware Q-former design.
> > * Clarifications on evaluation significance and fairness of baseline comparisons.
> > * Expanded discussion in the Related Work section, including a new subsection comparing prior Q-former–based approaches.
> >
> > As the discussion deadline is approaching, please let us know if anything remains unclear; we are on standby to clarify. Many thanks for your time.

---

### Official Review · Reviewer_bkBU · 2025-11-02

**Soundness:** 3
**Presentation:** 4
**Contribution:** 3
**Rating:** 6
**Confidence:** 4

**Summary:**

The paper proposes a multi-modal framework for Alzheimer’s Disease (AD) diagnosis that integrates several uni-modal foundation models (for sMRI, fMRI, clinical records, and genetic data). The core innovation is the modality-anchored interaction mechanism, where one modality serves as an anchor, and others act as auxiliary inputs. To align heterogeneous feature spaces, the authors introduce modality-aware Q-formers, transformer-based connectors that selectively map auxiliary features into the anchor modality’s feature space. The method is evaluated on ADNI, OASIS-3, and PPMI datasets for AD and Parkinson’s disease, showing state-of-the-art accuracy and generalization under both modality-complete and modality-incomplete scenarios.

**Strengths:**

1. Innovative framework: The modality-anchored interaction and modality-aware Q-former are novel and conceptually strong for integrating pre-trained models across heterogeneous medical modalities.
2. Strong performance: The proposed method consistently outperforms baselines across tasks, achieving superior accuracy, specificity, and sensitivity in AD diagnosis and progression prediction.
3. Comprehensive evaluation: Results are validated on multiple datasets (ADNI, OASIS-3, PPMI), demonstrating solid generalization to out-of-distribution data and to a different disease.
4. Thorough experimentation: Includes modality-complete/incomplete settings, ablation on foundation model choice, number of queries, and visualization of attention maps for interpretability.

**Weaknesses:**

1. Limited interpretability for clinicians: While attention visualization is provided, the interpretability of the fused multimodal decision process remains abstract; more clinical linkage to specific biomarkers would be valuable.
2. Complexity and scalability: The two-stage fine-tuning pipeline increases computational demands; practical feasibility in clinical deployment is not analyzed.
3. Dependence on large pre-trained models: The framework assumes availability of powerful foundation models, which may not always be accessible or easy to fine-tune with limited resources.
4. Limited cross-modal interpretive analysis: The paper would benefit from a deeper analysis of why certain modality combinations improve results (e.g., specific complementarity between clinical and imaging data).

**Questions:**

1. Q-former Configuration: How sensitive are the results to the number of queries and attention heads in the modality-aware Q-former? Could a lighter version achieve comparable performance?
2. Scalability: Can the approach be generalized to more than four modalities or to non-neuroimaging domains?
3. Interpretability: Can the method mechanisms be linked to specific neurobiological or genetic markers for clinical interpretability?

---

> ### Author Response · Authors · 2025-11-21
> **Rebuttal by Authors**
>
> # To Reviewer 1 (bkBU)
>
> We thank you for your constructive suggestions and appreciate comments like “Innovative framework”, “novel and conceptually strong”, “achieving superior accuracy”,  “comprehensive evaluation”, “solid generalization”, and “thorough experimentation”. We will gladly incorporate all the feedback in the revised version.
>
> ## Interpretability and Mechanism Linkage to Biomarkers
> > Q: Limited interpretability for clinicians: While attention visualization is provided, the interpretability of the fused multimodal decision process remains abstract; more clinical linkage to specific biomarkers would be valuable.
>
> > Q: Interpretability: Can the method mechanisms be linked to specific neurobiological or genetic markers for clinical interpretability?
>
> We appreciate the reviewer’s question regarding linking our fused multimodal decision process and mechanisms to  neurobiological and genetic markers for clinical interpretability. We would like to clarify that the attention mechanisms in our proposed Modality-aware Q-former is able to highlight clinically meaningful features, and these can be directly associated with established biomarkers for Alzheimer’s Disease (AD).
> ### Attention Visualization for Biomarker Associations
> As shown in the attention visualizations in Appendix D.5 (Fig. 10), we analyzed how the model allocates attention in those modalities to understand which biological and clinical signals influence the predictions. These patterns reveal clear associations with established AD biomarkers:
> * **Genetic and clinical marker**: As shown in Appendix D.5 (Fig. 10a), the model assigns high attention to well-established AD-related genetic biomarkers such as APOE (Knopman et al.,2007)[1,2,3], and emphasizes clinically relevant variables including MMSE scores, delayed recall tests, neuropsychological assessments, and medical history features. These variables are widely recognized as critical indicators in prior AD research [4].
> * **Neuroimaging markers**: In Fig. 10b, attention maps for sMRI consistently focus on regions known as core AD biomarkers—hippocampal atrophy, medial temporal lobe degeneration, and cortical thinning in parietal and temporal lobes[5,6,9]. Notably, the entorhinal cortex and hippocampal formation, among the most affected regions in AD, receive strong attention, closely matching established neurobiological findings [7,8,10].
>
> ### Multi-Modal Biomarker Associations via Q-Former Mechanisms
> From a structural perspective, our modality-anchored Q-formers are explicitly designed to enable information flow from relevant auxiliary modalities to the anchor modality. It allows the model to capture inter-modal relationships, meaning that it is capable of reflecting the biologically meaningful associations that exist between different modalities’ AD biomarkers.
>
> Experimentally, we visualize the attention maps of AD patients’ sMRI scans across different anchor-modality models trained in Stage 2 (BrainMVP and NT-Transformer), highlighting the brain regions that each model attends to most during prediction. The corresponding visualizations are provided in the revised version in Appendix D.6, Fig. 12. Specifically, the NT-Transformer trained with genetic features (including APOE ε4 status) can  highlight hippocampal and medial temporal lobe regions on sMRI. This pattern aligns closely with well-established AD neurobiology: extensive prior studies [11,12] have shown that APOE ε4 is strongly associated with hippocampal and parahippocampal atrophy. Therefore, the fact that our gene-anchored model also attends to hippocampal regions on sMRI indicates that the q-former successfully captures meaningful cross-modal biomarker relationships. This provides experimental evidence that our model learns biologically grounded correspondences between different modalities.
>
> Together with fMRI visualizations in Fig. 4, these results demonstrate that our learned representations align with known neurobiological and genetic markers, providing clinically interpretable evidence for the model’s decision process.

---

> > ### Author Response · Authors · 2025-11-21
> > **Rebuttal by Authors**
> >
> > ## Complexity and Scalability
> > > Q: Complexity and scalability: The two-stage fine-tuning pipeline increases computational demands; practical feasibility in clinical deployment is not analyzed.
> >
> > We appreciate your concern regarding computational demands. The two-stage fine-tuning pipeline impacts only the training phase. Inference with the final model is lightweight: it requires only 381 ms per subject on a single A100 GPU, with 23.4 GB (23,416 MB) of GPU memory consumption. This computational requirement remains well within clinically acceptable limits, ensuring practical feasibility for real-world deployment.
> >
> > Training of our two-stage pipeline requires 30 A100-GPU hours, with a training throughput of 6.85 s/sample. Due to LoRA, this single training run is sufficient to reach the reported performance. Since training is a one-time process and does not need to be repeated for deployment, a total of 30 GPU hours is practically acceptable in clinical settings.
> >
> > ## Availability and fine-tuning efficiency of models
> > > Q: Dependence on large pre-trained models: The framework assumes availability of powerful foundation models, which may not always be accessible or easy to fine-tune with limited resources.
> >
> > We appreciate the concern regarding reliance on large pre-trained models. In practice, most medical domains (e.g., MRI, CT, genetic data) now have widely accessible foundation models (e.g., BrainMVP (Rui et al.,2025), CT-FM[13], NT (Dalla-Torre et al.,2025), enabled by recent releases and large-scale datasets. Consequently, obtaining a strong pre-trained model is generally feasible in typical clinical settings.
> >
> > Importantly, our framework is designed to adapt when foundation models are unavailable for certain modalities. In such cases, the output of a foundation model can be replaced with predictions from any alternative model without altering the overall architecture. To evaluate robustness under this scenario, we conducted experiments using three modalities with foundation models (text, sMRI, fMRI), while the genetic modality was trained entirely from scratch using SEI(Chen et al.,2022).
> >
> > | **Method**       | **NC vs. MCI** | **NC vs. AD** | **sMCI vs. pMCI** |
> > |-------------------|---------------|--------------|-------------------|
> > | w/o Gene Data     | 0.930         | 0.912        | 0.771             |
> > | w/ SEI (from scratch) | 0.936         | 0.929        | 0.801             |
> > | **w/ NT**           | **0.979**         | **0.945**        | **0.846**             |
> >
> >
> > These results show that our framework remains effective even when a modality lacks a foundation model. Training SEI from scratch still provides clear benefits compared to removing the genetic modality entirely, demonstrating practical applicability in resource-limited scenarios. Moreover, with LoRA-based adaptation, the model can be fine-tuned even on less powerful hardware. While our experiments were run on A100 GPUs, we practically confirm that training remains feasible on a single A6000 GPU (70 A6000-GPU hours), further underscoring the method’s applicability under constrained computational resources.
> >
> > ## Cross-modal interpretive analysis
> > > Q: Limited cross-modal interpretive analysis: The paper would benefit from a deeper analysis of why certain modality combinations improve results (e.g., specific complementarity between clinical and imaging data).
> >
> > We thank the reviewer for the insightful suggestion to provide a deeper interpretive analysis of modality combinations. Following this recommendation, we revisited the results in Figure 3 and analyzed why certain combinations yield the most pronounced gains. Our exhaustive ablation study shows that clinical records and fMRI deliver the strongest improvement when fused. This synergy stems from their intrinsic properties:
> >
> >  * Clinical assessments capture high-level behavioral and cognitive impairments—such as memory decline, language deficits, and executive dysfunction—that are not directly observable in fMRI signals. These impairments are reflected through neuropsychological tests and cognitive scores.
> >  * fMRI, on the other hand, reveals neural activity patterns, regional activation deficits, and network-level dysfunction—such as disrupted connectivity in the default mode network or abnormal activation in hippocampal regions—that clinical data cannot detect.
> >
> > These non-overlapping cues provide complementary perspectives: clinical data offers systemic indicators of functional decline, whereas fMRI provides fine-grained neurophysiological evidence. The integration of these heterogeneous signals naturally enhances predictive power, explaining the performance improvements observed in Figure 3. This interpretive analysis confirms that the observed gains are grounded in the complementary nature of these modalities rather than incidental correlations.

---

> ### Author Response · Authors · 2025-11-21
> **Rebuttal by Authors**
>
> ## Q-former Configuration
> > Q: Q-former Configuration: How sensitive are the results to the number of queries and attention heads in the modality-aware Q-former? Could a lighter version achieve comparable performance?
>
> We appreciate the reviewer’s question regarding the sensitivity of our framework to Q-former configurations. As shown in Figure 5 of the original submission, we previously evaluated the impact of query numbers, where 16 queries yielded the best performance. To further address this concern, we conducted an extended sweep with queries ∈ {0, 4, 8, 16, 32}:
>
>
> | **Query Number**       | **0**   | **4**   | **8**   | **16**  | **32**  |
> |------------------|--------|--------|--------|--------|--------|
> | **NC vs. MCI**   | 0.881  | 0.903  | 0.956  | **0.971**  | 0.969  |
> | **NC vs. AD**    | 0.899  | 0.913  | 0.935  | **0.941**  | 0.940  |
>
> These results show that a lighter 8-query configuration achieves performance very close to the optimal 16-query setting, with only a marginal drop in accuracy.
>
> We also swept attention heads ∈ {3, 6, 12, 24}, keeping other settings constant:
>
>
> | **Attention Heads**       | **3**   | **6**   | **12**  | **24**  |
> |------------------|--------|--------|--------|--------|
> | **NC vs. MCI**   | 0.959  | 0.965  | 0.971  | 0.973  |
> | **NC vs. AD**    | 0.934  | 0.937  | 0.941  | 0.942  |
>
>
> Although accuracy slightly improves with more heads, overall performance remains stable across configurations. This demonstrates that the Modality-aware Q-former is not highly sensitive to these hyperparameters, and lighter versions can achieve comparable performance. Therefore, our design offers flexibility for resource-constrained settings without sacrificing accuracy.
>
> ## Scalability of Modalities and Domains
> > Q: Scalability: Can the approach be generalized to more than four modalities or to non-neuroimaging domains?
>
> We thank the reviewer for raising the question on scalability. Our framework is inherently flexible with respect to the number and type of modalities. It does not impose architectural constraints tied to a fixed set of inputs, allowing seamless integration of additional modalities beyond the four used in our experiments.
>
> Importantly, the current implementation already includes non-neuroimaging data, clinical records and genetic data, demonstrating its applicability beyond imaging. The same design principles enable straightforward extension to other non-neuro domains.
>
> | **Modality**        | **Methods**     | **C-Index**         |
> |-----------------|-------------|-----------------|
> | CT              | CT-FM       | 0.723 ± 0.013   |
> | Textual         | LLaMA       | 0.697 ± 0.027   |
> | CT + Textual    | lateFusion  | 0.730 ± 0.011   |
> | **CT + Textual**    | **Ours**        | **0.776 ± 0.020**   |
>
>
> To demonstrate that our approach can generalize beyond neuroimaging, we further evaluated it on the TCGA Kidney survival prediction task using CT imaging (CT-FM [13]) and clinical text data (LLaMA). Following prior survival-prediction settings and using the C-index for evaluation [14], our method again achieved strong performance, showing that it can be effectively extended to additional medical modalities and non-neuro domains.

---

> ### Author Response · Authors · 2025-11-21
> **Rebuttal by Authors**
>
> ## Reference
> [1]Kim, J et al., (2009). The role of apolipoprotein E in Alzheimer's disease. Neuron, 63(3), 287-303.
>
> [2]Angelopoulou, E et al., (2021). APOE genotype and Alzheimer’s disease: the influence of lifestyle and environmental factors. ACS chemical neuroscience.
>
> [3]Stocker, H et al., (2018). The genetic risk of Alzheimer’s disease beyond APOE ε4: systematic review of Alzheimer’s genetic risk scores. Translational Psychiatry, 8(1), 166.
>
> [4]Dubois, Bruno, et al. "Biomarkers in Alzheimer’s disease: role in early and differential diagnosis and recognition of atypical variants." Alzheimer's Research & Therapy 15.1 (2023): 175.
>
> [5]Falgàs, N et al., (2019). Hippocampal atrophy has limited usefulness as a diagnostic biomarker on the early onset Alzheimer's disease patients: a comparison between visual and quantitative assessment. NeuroImage: Clinical, 23, 101927.
>
> [6]Feng, F et al., (2021). Altered volume and structural connectivity of the hippocampus in Alzheimer’s disease and amnestic mild cognitive impairment. Frontiers in Aging Neuroscience, 13, 705030.
>
> [7]Woodward et al., (2024). The relationship between hippocampal changes in healthy aging and Alzheimer’s disease: a systematic literature review. Frontiers in aging neuroscience, 16, 1390574.
>
> [8]Kulason et al., Alzheimer's Disease Neuroimaging Initiative. (2019). Cortical thickness atrophy in the transentorhinal cortex in mild cognitive impairment. NeuroImage: Clinical, 21, 101617.
>
> [9]Zhao et al., (2015). Atrophic patterns of the frontal-subcortical circuits in patients with mild cognitive impairment and Alzheimer’s disease. PloS one, 10(6), e0130017.
>
> [10]da Silva Filho, S. R. B et al., (2017). Neuro-degeneration profile of Alzheimer's patients: a brain morphometry study. NeuroImage: Clinical, 15, 15-24.
>
> [11]Bailey et al., (2024). Impact of apolipoprotein E ε4 in Alzheimer’s disease: a meta-analysis of voxel-based morphometry studies. Journal of clinical neurology (Seoul, Korea), 20(5), 469.
>
> [12]Li et al., Alzheimer’s Disease Neuroimaging Initiative. (2016). Influence of APOE genotype on hippocampal atrophy over time-an N= 1925 surface-based ADNI study. PloS one, 11(4), e0152901.
>
> [13]Pai, et al. "Vision foundation models for computed tomography." arXiv (2025).
>
> [14]Jaume, et al. "Modeling dense multimodal interactions between biological pathways and histology for survival prediction." CVPR 2024.

---

### Author Response · Authors · 2025-12-02
**Rebuttal Summary for AC**

We provide a summary of our rebuttal here for AC to better understand the key aspects of our response. During the rebuttal, we carefully addressed every concern with substantial new experiments and clarifications. Overall, two reviewers gave positive scores (bkBU, score 6; gHU6, score 8), and we sincerely thank them for their encouraging comments, highlighting our work as “innovative”, “conceptually strong”, “solid”, “promising”, "novel and interesting", and demonstrating “superior accuracy”, “strong generalization”, and “thorough experimentation”. In the rebuttal and revised version, we addressed Reviewer bkBU's concern about interpretability by adding clearer biomarker-linked analyses and visualization results (Appendix D.5 Fig. 10; D.6 Fig. 12; main-text Fig. 4). We also addressed Reviewer gHU6’s concern regarding reproducibility and complexity. While the original submission already provided code and full setup details (Appendix B.1, B.2), in the rebuttal we additionally reported clear computational costs for training and inference to demonstrate practical feasibility.

We thank Reviewer x1Jq for the insightful comments. In our rebuttal, we directly addressed all three concerns:
* **Generalizability beyond classification.** We conducted experiments to extend our method to regression tasks and even non-neuro setting (TCGA survival prediction).
 * **Technical novelty of Q-former.** We clarified with concrete evidence that our modality-aware Q-former is an innovative mechanism, architecturally and functionally distinct from prior Q-former designs.
 * **Statistical significance.** We supplemented our existing significance analysis in Appendix B.1 by adding standard deviation reporting for all baselines in our response.

We thank Reviewer okfp for the comments. We respectfully note that two of the concerns were based on misunderstandings of our work, and we would like to clarify them directly.
* **Misunderstanding about Modality-aware Q-formers.** The reviewer claimed that "several existing works are about Modality-aware Q-formers (Tong 24; Zong 24; Alayrac 22; Liu 23)", but none of these cited works use "Q-formers" at all—these methods rely on connectors rather than query transformers. The concept of a *Modality-aware Q-former* is introduced for the first time in our work.
* **Misunderstanding about the breadth of baseline comparisons.** The reviewer claimed our baseline comparisons were limited, which is incorrect. Our original submission already included 17 SOTA AD diagnosis baselines spanning uni-, bi-, and multi-modal methods, making it one of the most comprehensive comparisons in this area.

Furthermore, in response to Reviewer okfp, we clarified that our modality-aware Q-former introduces genuine architectural and functional innovations, and that beyond this technically innovative module, the proposed modality-anchored interaction constitutes a novel fusion framework that is fundamentally different from prior multimodal methods.

We sincerely appreciate AC's precious time and thoughtful attention to our work, and we kindly ask AC to consider our response and rebuttal during assessment.

---

### Meta-Review · Area_Chair_eurD · 2026-01-08

**Summary:**

This paper proposes a multi-modal framework for Alzheimer's Disease diagnosis that integrates four modalities (sMRI, fMRI, clinical records, and genetic data) through modality-anchored interaction and modality-aware Q-formers.

Key strengths identified across reviews:
1. Novel framework design with strong conceptual foundation
2. Comprehensive experimental validation across multiple datasets
3. Superior performance compared to baselines
4. Strong generalization to external datasets and other diseases

Main concerns raised:
1. Technical novelty (Reviewers x1Jq, okfp): Questions about the distinctiveness of the modality-aware Q-former compared to existing Q-former architectures
2. Interpretability (Reviewer bkBU): Limited clinical linkage to specific biomarkers
3. Complexity and reproducibility (Reviewers bkBU, gHU6): Concerns about computational demands and practical deployment
4. Scope of evaluation (Reviewer x1Jq): Focus primarily on classification tasks

**Reviewer Concerns:**

Effectively Addressed:
Interpretability: Authors provided comprehensive biomarker linkage analysis (Appendix D.5, D.6) showing attention to APOE genes, hippocampal regions, and other clinically relevant features
Computational complexity: Clarified inference time (381ms/subject) and training cost (30 A100-GPU hours), demonstrating practical feasibility
Q-former configuration sensitivity: Extended ablation showing robustness across query numbers and attention heads
Scalability: Demonstrated extension to TCGA kidney survival prediction (non-neuro domain)

Outstanding Concerns:

Technical novelty of framework: Authors provided detailed explanation of three novelty aspects (breadth of modality inclusivity, asymmetric fusion, novel Q-former design), but reviewer's fundamental concern about limited novelty may persist
Modality-aware Q-former distinction: Authors correctly identified that cited papers (Tong et al., Zong et al., Alayrac et al., Liu et al.) do NOT use Q-formers but rather connectors. However, this reveals the reviewer may have misunderstood the related work
While computational costs are now documented, practical deployment in resource-constrained clinical settings remains somewhat uncertain. The two-stage training pipeline, while justified, still adds complexity that may limit adoption

**Reviewer Scores:**

The paper demonstrates strong empirical results, comprehensive experiments, and addresses an important clinical problem. The rebuttal was exceptionally thorough and addressed nearly all concerns with substantial new experiments. The main points supporting acceptance:
1. Strong empirical validation: Consistent improvements across multiple datasets, tasks, and diseases
2. Technical contributions: Despite debates about novelty, the modality-anchored interaction and modality-aware Q-former represent meaningful advances for multi-modal medical diagnosis
3. Comprehensive evaluation: Authors went beyond original experiments to address reviewer concerns
4. Clinical relevance: Strong biomarker alignment and interpretability

---

### Decision · Program_Chairs · 2026-01-26

Accept (Poster)